ecology, environmental science, evolution

invasive species, introgression, naturalization, microsatellite, pig, boar

**Authors for correspondence:**
Donovan Anderson
e-mail: donovananderson2018@outlook.com
Shingo Kaneko
e-mail: kaneko.shingo@gmail.com

†Present address: Center for Research in Isotopes and Environmental Dynamics, University of Tsukuba, Tsukuba, Ibaraki, Japan.

# Introgression dynamics from invasive pigs into wild boar following the March 2011 natural and anthropogenic disasters at Fukushima

Donovan Anderson[1,†], Yuki Negishi[1], Hiroko Ishiniwa[2], Kei Okuda[3], Thomas G. Hinton[4], Rio Toma[1], Junco Nagata[5], Hidetoshi B. Tamate[6] and Shingo Kaneko[1,2]

[1]Symbiotic Systems Science and Technology, Fukushima University, Fukushima City, Fukushima, Japan
[2]Institute of Environmental Radioactivity, Fukushima University, Fukushima City, Fukushima, Japan
[3]Faculty of Human Environmental Studies, Hiroshima Shudo University, Hiroshima, Hiroshima, Japan
[4]Centre for Environmental Radioactivity, Faculty of Environmental Sciences and Natural Resource Management, Norwegian University of Life Sciences, Ås, Norway
[5]Forestry and Forest Products Research Institute, Tsukuba, Ibaraki, Japan
[6]Yamagata University, Yamagata City, Yamagata, Japan

  DA, 0000-0001-6716-4059; HI, 0000-0001-7712-8051; KO, 0000-0001-7997-0701; JN, 0000-0001-9182-319X; HBT, 0000-0001-5255-3350; SK, 0000-0002-9021-8155

Natural and anthropogenic disasters have the capability to cause sudden extrinsic environmental changes and long-lasting perturbations including invasive species, species expansion and influence evolution as selective pressures force adaption. Such disasters occurred on 11 March 2011, in Fukushima, Japan, when an earthquake, tsunami and meltdown of a nuclear power plant all drastically reformed anthropogenic land use. Using genetic data, we demonstrate how wild boar (*Sus scrofa leucomystax*) have persevered against these environmental changes, including an invasion of escaped domestic pigs (*Sus scrofa domesticus*). Concurrently, we show evidence of successful hybridization between pigs and native wild boar in this area; however in future offspring, the pig legacy has been diluted through time. We speculate that the range expansion dynamics inhibit long-term introgression and introgressed alleles will continue to decrease at each generation while only maternally inherited organelles will persist. Using the gene flow data among wild boar, we assume that offspring from hybrid lineages will continue dispersal north at low frequencies as climates warm. We conclude that future risks for wild boar in this area include intraspecies competition, revitalization of human-related disruptions and disease outbreaks.

## 1. Introduction

Anthropogenic land use can cause ecological impacts to animals inhabiting the landscape, such as limiting range expansion, driving some species to extinction and overall loss of biodiversity [1,2]. Drastic shifts in land use occur as a result of anthropogenic or natural disasters; and these changes are often coupled with new anthropogenic pressures, such as escaped invasive species [3], habitat disruptions [4] and, in rare incidents, an influx of radioactivity contamination [5]. However, while sudden environmental changes or new anthropogenic pressures have caused severe reduction in biodiversity and extinction [6,7], ecosystems have demonstrated ecological resilience and sometimes animal populations experience greater dispersal and increased abundance [8,9]. Such extrinsic ecological changes could cause long-lasting perturbations and challenges, including invasive species,

fitness declines and sudden fluctuations in population densities, which may considerably influence evolution.

Radioactive materials were dispersed as a result from both the Chernobyl and Fukushima Dai-ichi Nuclear Power Plant (CNPP and FDNPP) accidents in 1986 and 2011, respectively [5], causing widespread radiological contamination of the environments and chronic exposure to ionizing radiation. Consequently, both national governments mandated an evacuation of humans from large areas surrounding the nuclear accident sites (i.e. 4300 km$^2$ for the CNPP and 1150 km$^2$ for the FDNPP). In the evacuated areas, wildlife census studies have demonstrated a rewilding occurred over time, with an abundance of large mammals in both these ecological communities [10–12]. Increased animal abundance of various species in the evacuated area might cause new pressures, such as overpopulation and resource depletion [9]. In the Chernobyl evacuated area, wild boar (*Sus scrofa*) saw a drastic population increase, but it was stabilized and even declined as predatory populations and disease increased [11,12]. Wild boar in the Fukushima evacuation area could be experiencing changes in population size and behaviour, as suggested by an increased abundance and shifts to more diurnal behaviour [10]; however, these populations lack predators to counter the population increase, other than government-sponsored culling programmes. Evaluation of genetic data and gene flow among populations with indication of expansion trends and high population densities, presumably as a result of sudden ecological changes or a rewilding, may shed light on the adaptive process.

Human abandonment of such a large area at Fukushima may have provided favourable conditions for a rapid increase in those wildlife species that were able to benefit from landscapes that were formally anthropocentric [10]. Concurrently, the natural disasters and forced abandonment of farming communities at Fukushima resulted in the release of domesticated livestock into the same landscapes, and it is known that escaped porcine livestock can naturalize in the wild and breed with their wild relatives [13,14]. The favourable environments established new invasive species interactions with the pre-existing species, resulting in intraspecies hybridization, which may alter genetic traits and genomes of the native species [15,16]. The interactions and genetic mixing from intraspecies hybridization can be used to provide key insight on the naturalization, range expansion dynamics and natural selection phenomena of influenced species in these landscapes. Additionally, any introgressed alleles or organelles in the native population that increase, protract or decrease over a period of time can be used to determine if such invasions from a suddenly introduced population (i.e. release of livestock following a natural disaster or forced evacuations) have ecological benefits.

We focus on wild boar in Fukushima's evacuated area because wild boar here are experiencing: (i) sudden population expansion as evidenced by greater population abundance [10] and population growth in the area (estimated 49 000 to 62 000 wild boar from 2014 to 2018 [17]); (ii) reduced anthropogenic disturbances because of human evacuations [4], which included approximately 300 km$^2$ of urban and agricultural lands [5]; and (iii) challenges from a recent invasion of about 30 000 pigs (*Sus scrofa domesticus*), released from abandoned farmlands [18] causing population intraspecies hybridization (see phenotypical colour alteration in electronic supplementary material, figure S1). Presumably, the native wild boar

populations may benefit from an introgression of pigs, known for high genetic diversity, novel genotypes and heterosis [13,14,19], which may enhance the chances of adaptation and expansion [20]. However, intraspecies hybridization may also lead to the replacement of the native species by the invading species [21], may cause maladaptation to the natural environment [22] and specifically, intraspecies hybridization between pigs and wild boar may cause alterations of genetic traits including litter sizes, immunology and population expansion dynamics [23]. Thus, investigating intraspecies hybridization and introgression in a natural population influenced by a recent large invasion could infer important information on the complex histories of hybrids including selection, fitness in hybrids or the invaders, and admixture dynamics from range expansion.

Here, we evaluate the hypotheses that range expansion acts as resistance against hybridization, and that invasion success is dependent on the invaders' response to naturalization; both hypotheses have been often embedded in evolution [22,24,25]. Specifically, we determined the extent of the genetic introgression from invasive pigs into wild boar in Fukushima Prefecture after the FDNPP accident, and observed the longevity of introgressed alleles or organelles to evaluate the above hypotheses. In addition, we were able to predict the initial contact zone for pigs and wild boar and evaluate the dispersal of offspring from hybrid lineages using genetic data. We show that if future hybridization events occur, they will occur at low frequencies and that the introgressed pig genes should continue to dilute along the expansion of hybrids. Such results would provide evidence of demic diffusion in the wild and that intraspecies competition or lack of survivability of the invasive species has prevented the invaders' naturalization.

## 2. Material and methods

### (a) Study area

Due to high-dose rates from radioactive material dispersed in March 2011, the Japanese Government issued an evacuation order for 164 845 people living within a 20 km radius of the FDNPP (https://www.pref.fukushima.lg.jp/site/portal/). The evacuation order was subsequently modified to include areas as far as 40 km northwest of the FDNPP that were also contaminated by the radioactive plume, comprising a total area of approximately 1150 km$^2$ [5,10]. Beginning in 2016, some of the evacuated areas have been remediated and a small percentage of the former residents have returned; however, human access remains severely restricted in the most contaminated areas as a result of the radiation exposure exceeding the safety threshold for humans set by the Japanese Government. The reduced human activities and altered anthropogenic pressures created favourable habitat conditions for mid- to large mammal species, such as wild boar, as evidenced by increased densities in the evacuated area [10].

### (b) Sampling and DNA extraction

Muscle samples were collected from 191 wild boar, all of which were morphologically identified as typical wild boar in Japan [26], captured in or nearby the Fukushima evacuated zone (described above) from 2015 to 2018 (figure 1). Additional wild boar samples from the period prior to the 2011 evacuations were provided by prefectural hunters, which included 25 muscle samples from a wild boar population in Ibaraki Prefecture, south of Fukushima Prefecture; 10 muscle samples from Yamagata

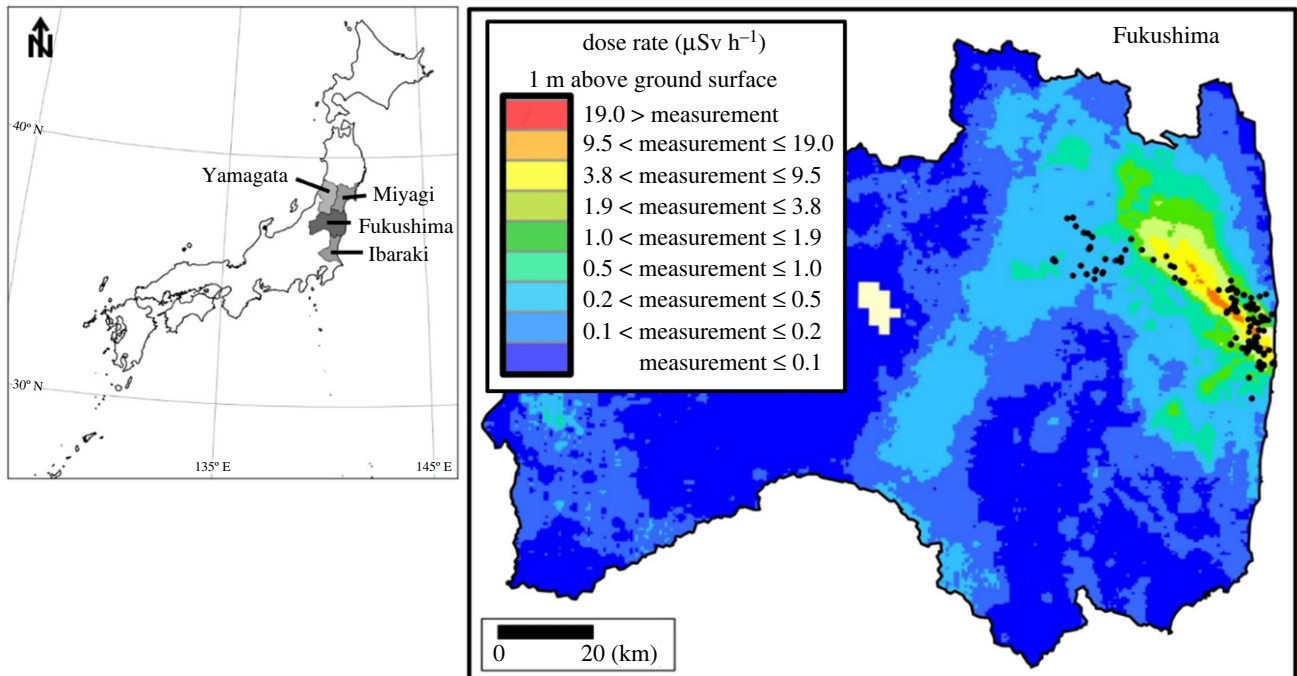

**Figure 1.** Distribution map of wild boar samples collected in the evacuated zone and nearby areas impacted by radiation dispersal from the Fukushima Dai-ichi Nuclear Power Plant accident in 2011. Other samples were collected from Yamagata, Miyagi and Ibaraki Prefectures, which neighbour Fukushima and are indicated in the inset map. Ambient dose rate ($\mu$Sv h$^{-1}$) measurements (1 m above ground surface) are shown for November 2016, provided by the Ministry of Education, Culture, Sports, Science and Technology (MEXT) and Nuclear Regulation Authority (NSR) airborne monitoring project. Map is sourced from Extension Site of Distribution Map of Radiation Dose (MEXT/NSR) site (https://ramap.jmc.or.jp/map). (Online version in colour.)

Prefecture and seven muscle samples from Miyagi Prefecture, both north of Fukushima Prefecture. GPS coordinates, sex and estimated age based on tooth erosion patterns were recorded at the trap sites prior to sampling. See electronic supplementary material for approximating the birth period using sampling year and age data. Ten pig muscle samples were obtained from a Fukushima pig slaughterhouse or purchased from a Fukushima prefectural meat market in 2016. In total, 243 samples were assayed for this study. Among the samples, the samples from Fukushima, Miyagi and Yamagata Prefectures, and 10 pig samples were previously used in Anderson *et al.* [27]; and the 25 samples from Ibaraki Prefecture were used in Nagata *et al.* [28], which both analysed the mitochondrial DNA (mtDNA) control region. All samples were stored individually at −20°C in 99.5% ethanol until DNA extraction. Total genomic DNA was extracted using the Gentra PureGene Blood & Tissue kit (QIAGEN, CA, USA), according to the manufacturer's instructions.

## (c) Genetic analysis

### (i) Mitochondrial DNA
The mtDNA sequences provide material lineage information and can be used to identify wild boar with pig maternal ancestry for many distant generations. The mtDNA sequences were obtained from previous literature for all samples in this study [27,28] . In brief, the mtDNA control region was successfully amplified and partial sequences (713 bp) were determined from all samples. DNA sequencing data were viewed from FinchTV chromatogram viewer v. 1.5.0 (Geospiza Inc., WA, USA).

### (ii) Nuclear microsatellites
Nuclear microsatellite (STR) data were analysed to determine the genetic structure of wild boar and the extent of domestic pig introgression into the local wild boar population. A total of 24 STR loci were selected and genotyped for our study populations based on the allele frequencies and the amplification for each of these

markers in pure species individuals (see Anderson *et al.* [29]). All 24 markers were developed by previous studies [30–32] and recommended by the Food and Agriculture Organisation of the United Nations database [33]. For 31 samples already analysed in Anderson *et al.* [29], we used the genotypes reported therein for the 24 STR loci. The remaining 212 samples were analysed as described in Anderson *et al.* [29]. In brief, PCR amplification was performed in 5 µl reactions using the QIAGEN Multiplex PCR Kit (QIAGEN, CA, USA) and a protocol for fluorescent labelling [34]. Each sample reaction contained 10–20 ng of genomic template DNA, 2.5 µl of Multiplex PCR Master Mix, 0.1 µM of forward primer, 0.2 µM of reverse primer and 0.1 µM of fluorescently labelled primer. Product sizes were determined using an ABI PRISM 3130 Genetic Analyzer and GeneMapper software (Applied Biosystems, MA, USA).

## (d) Data analyses
Our genetic data, both mtDNA and STR, were divided into two groups for analyses: (i) individual-based data analyses to assess introgression from pigs into wild boar, i.e. intraspecies hybridization in the evacuation zone or nearby areas. Samples from the period prior to the 2011 evacuations and outside Fukushima Prefecture were excluded from this group. (ii) Population-based data analyses to predict possible dispersal patterns of hybrids using gene flow and genetic structure of the native wild boar populations within the evacuated area, to the north (Miyagi and Yamagata Prefectures) and to the south (Ibaraki Prefecture) of the evacuation zone.

### (i) Hybrid analyses of crossed wild boar and pigs
The STR genetic structures and genomic mixtures between wild boar and pigs were investigated using a Bayesian clustering algorithm available in STRUCTURE v. 2.3.4 [35,36]. STRUCTURE was run using the settings of the admixture model with correlated allele frequencies; all parameters were set to default with the LOC-PRIOR model implemented. To estimate the number of clusters

**Figure 2.** Results from genetic clustering conducted using STRUCTURE analysis ($K = 2$) based on 24 loci for hybrid analysis. The membership bar-plot indicates per cent membership to pig, where values less than 1% indicated pure-wild boar ancestry and greater than 99% indicated pig ancestry. 'Introgressed' indicates admixed individuals. The morphology information was determined in the field from Anderson *et al.* [27]. Mitochondria results of 'pig' indicated the typical pig haplotype, and 'boar' indicated typical Japanese wild boar haplotypes.

($K$), 10 independent runs with $K = 1$–10 were performed using $10^6$ Markov chain Monte Carlo (MCMC) iterations following a burn-in of $10^5$ iterations. STRUCTURE HARVESTER [37] was used to calculate the probability of the data for each $K$ (Ln$P(D)$) and $\Delta K$ [38]. The Ln$P(D)$ remained at 0 with each $K$ except a high peak at $K = 2$ (electronic supplementary material, figure S2), and the highest $\Delta K$ was detected when $K = 2$. Taken together, $K = 2$ was the optimal number and we retained $K = 2$ to identify the proportions of admixture in wild boar and pig hybrids. For the selected $K = 2$, we assessed the average proportion of the membership coefficient ($Q_1$) to the inferred clusters. We assigned each individual to the inferred clusters, using a conservative threshold $Q_1 \geq 0.99$ for the assignment of individuals genomes to the pig cluster, or, $Q_1 < 0.01$ to the wild boar cluster. Admixed individuals were jointly assigned to the two clusters and were considered to have possible hybrid ancestry or were the offspring from ancestors that were a cross between wild boar and pig. Admixed individuals were considered hybrids only if neither of the thresholds were true in all independent runs. If an individual admixed in more than 50% of the runs, but not all, then the individual was considered as a 'suggested hybrid'.

For clarity, terms used in this manuscript are defined as follows. First, 'hybridization' refers to mixing between evolutionarily distinct lineages, whereas 'introgression' refers to gene flow between species as a consequence of hybridization. In this study, introgressed or hybrid wild boar were determined based on the inherited pig lineage haplotype from mtDNA, or an admixed individual based on STRUCTURE (see above), or both.

Frequencies of wild boar identified as hybrids were partitioned by the corresponding evacuation zones based on GPS coordinates: initial evacuation zone, extended evacuated zone and outside the evacuation zone. To geographically evaluate wild boar considered as hybrids based on either method, two maps were created using corresponding GPS coordinates and (i) colour based on haplotype or (ii) a colour gradient scale based on $Q_1$ values. Distances from wild boar trap sites and the FDNPP (37°25′23″ N, 141°01′59″ E) were calculated using the geographical distance-based analysis in GenALEx v. 6.41 [39]. We calculated pairwise codominant genotypic distances [40], and performing principal coordinates analysis (PCA) using GenALEx v. 6.41 [39].

### (ii) Population-based genetic analyses of wild boar

For the subsequent analyses, only individuals that had typical mtDNA haplotypes of wild boar, i.e. J10 and J5, and assigned to the inferred wild boar cluster were used to determine gene flow among the native wild boar populations in this region, as such information could help provide predictions on future dispersal. Samples used for these genetic analyses were considered as pure strain wild boar, or wild boar with only wild boar ancestry and lineages, and are henceforth referred to as 'pure-wild boar'. The STR data from 202 pure-wild boar, including 160 pure-wild boar from the evacuated zone and 42 pure-wild boar from the neighbouring prefectures to the north and south, were used.

Genetic structure of pure-wild boar was investigated by performing a STRUCTURE analysis and a test of isolation by distance. We ran STRUCTURE with the same setting parameters as discussed above and the same criteria were used for $K$ and $\Delta K$. We examined the association between the matrix of the geographical distances and Queller & Goodnights genetics relatedness ($r$) [41] with three groups: (i) pure-wild boar inside the evacuated zone, (ii) the northern cluster as identified by STRUCTURE, and (iii) among all pure-wild boar, using a Mantel test with 9999 random permutations in GenALEx v. 6.41 [39].

## 3. Results

### (a) Genetic analyses of hybridization in wild boar

*MtDNA*. Two typical wild boar haplotypes (J10 and J5; D42172 and AB015085, respectively) and one typical pig lineage haplotype (H1; MK801664) were observed in wild boar surveyed for mtDNA. For the 191 wild boar samples from within the evacuated zone, the number (and proportion) of each haplotype was 170 (89%), three (1.6%) and 18 (9.4%) for J10, J5 and H1, respectively. In other words, 173 (90.6%) of the mtDNA haplotypes were typical for wild boar lineages and 18 (9.4%) showed evidence of pig ancestry.

*Nuclear STR*. The STRUCTURE and PCA analysis used to investigate the introgression of pig alleles into native wild boar based on 24 loci gave similar genetic structuring compared to that obtained from mtDNA and distinguished wild boar, pig and hybridized individuals. The most probable number of clusters that captured greatest proportion of the data in the STRUCTURE analysis was $K = 2$ (figure 2) when the Ln$P(D)$ and the $\Delta K$ were evaluated (electronic supplementary material, figure S2). All individuals were assigned to two clusters at $K = 2$, a first group comprised the 10 pigs, and a second group comprised all the wild boar within the Fukushima evacuation area with 10 admixed individuals in 100% of the independent runs and an additional five admixed individuals in more than 50% of the runs. The estimated $Q_1$ values for all members of the first group were $Q_1 \geq 0.99$, while the second group consisted of 160 individuals with a $Q_1 < 0.01$. The 10

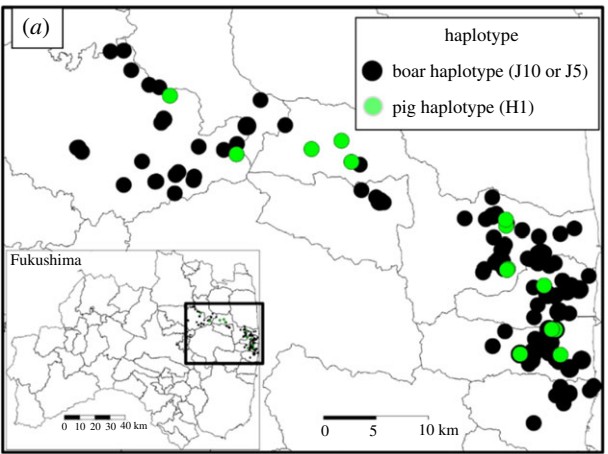
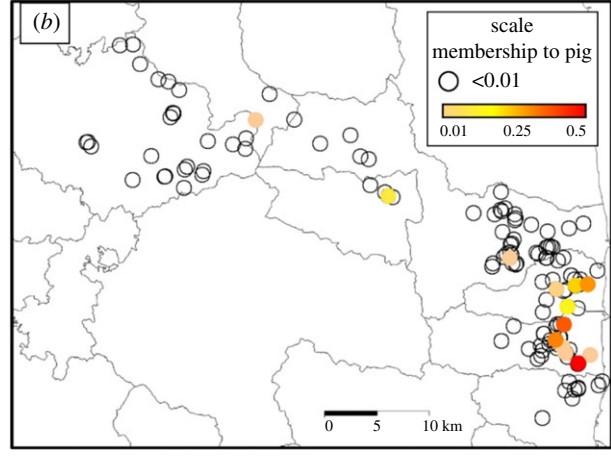

**Figure 3.** Distribution map of all wild boar samples in the evacuated area and either their (*a*) mtDNA haplotype data or (*b*) the estimated proportion of the membership coefficient ($Q_1$) obtained by STRUCTURE. (Online version in colour.)

**Table 1.** Number of hybrids detected using mtDNA and STR data from wild boar samples within the initial evacuation zone, extended evacuated area and outside the evacuation zone, which had 20, 20–40 and greater than 40 km radii, respectively, from the FDNPP. $Q_1$, estimated proportion of the membership coefficient by STRUCTURE.

| distance (km) | *n* | no. of hybrids detected | | | | $Q_1$ value of hybrids | |
| | | mtDNA | STR[a] | mtDNA and STR | total (per cent of all) | range | average |
|---|---|---|---|---|---|---|---|
| <20 | 149 | 11 | 11 (4) | 2 | 24 (16%) | 0.01–0.51 | 0.1 |
| 20–40 | 24 | 4 | 2 (1) | 0 | 6 (25%) | 0.01–0.11 | 0.02 |
| >40 | 18 | 1 | 0 | 0 | 1 (6%) | n.a. | n.a. |
| all | 191 | 16 | 13 (5) | 2 | 31 (16%) | 0.01–0.51 | 0.08 |

[a]No. of 'suggested hybrid' in parentheses (see Material and methods).

admixed individuals in 100% of the runs and additional five admixed individuals in more than 50% of the runs ranged from $0.03 \leq Q_1 \leq 0.51$ and $0.01 \leq Q_1 \leq 0.03$, respectively (figure 2; see more detail in electronic supplementary material, table S1). PCA analysis of the STR data showed similar genetic clustering and separated the dataset into the two genetic clusters, with the identified hybrid wild boar positioned separately from the wild boar cluster, trailing off in the direction of the pig cluster (electronic supplementary material, figure S3).

Combined mtDNA and STR results indicated that 31 wild boar, or 16% of the wild boar from the evacuated zone, were hybrids (table 1). The distribution map of wild boar in the evacuated area coupled with either the mtDNA information or $Q_1$ values (figure 3) revealed that the distribution of hybrids was analogous for each dataset and showed similar tendencies with distance from the FDNPP. The greatest number of hybrids detected, or relatively 75% of all hybrids, were within the initial 20 km radius evacuation zone (table 1). The shared ancestry to pig was higher for hybrids in the initial 20 km radius evacuation zone or, in other words, there was evidence of higher introgression inside the initial evacuated zone (figure 4). Outside the evacuation zone, only one female hybrid wild boar was identified using mtDNA (see electronic supplementary material for sex information and table S1). Using the estimated birth year of hybrid wild boars identified using STR analysis, five hybrids were born in 2016 that ranged from 0.03 to 0.51 (average = 0.27)

and five hybrids were born in 2018 that ranged from 0.04 to 0.33 (average = 0.21) (electronic supplementary material, table S1).

## (b) Genetic structure and admixture among pure-wild boar

Closely related populations and a separation of a southern and northern cluster among wild boar populations across our study's wide region, spanning four prefectures of Japan, were observed by the pure-wild boar STRUCTURE analysis. The Ln*P*(*D*) increased progressively from *K* = 2 to 3 and then plateaued after *K* = 3 (electronic supplementary material, figure S4). Additionally, the highest Δ*K* was detected when *K* = 2 and the second highest when *K* = 3. Thus, the most likely number of clusters was *K* = 2 followed by *K* = 3. At *K* = 2, the range of the wild boar was divided into two clusters (figure 5). A first group was detected with a shared ancestry in all populations and dominated the wild boar ancestry in Yamagata Prefecture, Miyagi Prefecture and the Fukushima evacuated zones. A second group was detected that dominated in Ibaraki Prefecture, with a decreasing share of ancestry moving north. At *K* = 3, a third cluster showed similar share of ancestry between northern prefectures, and the Fukushima evacuated zones, but drastically decreased moving south. The clustering results agreed with PCA results (electronic supplementary material, figure S5), which also indicated wild boar

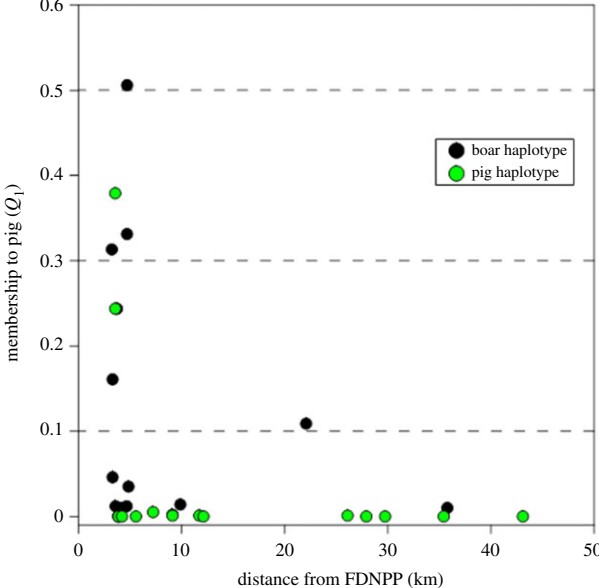

**Figure 4.** The estimated proportion of the membership coefficient ($Q_1$) plotted against the distance from the Fukushima Dai-ichi Nuclear Power Plant. Only identified hybrid wild boar were plotted. (Online version in colour.)

scattered across two genetic clusters. In other words, the Fukushima evacuated zone and northern prefectures' populations tended to overlap positions, while the southern Ibaraki population was positioned separately. In addition, the northern group wild boar collected from the period prior to 2011, suggested little genetic drift from the samples collected after 2011 ($F = 0.197$, figure 5), while the southern group samples diverged from the northern group (figure 5; see electronic supplementary material, figure S5).

Limited gene flow or mixing between the populations from the two main regions identified by STRUCTURE, the southern and northern separation, was also evident from the association between the matrix of the geographical distances and genetics relatedness analysis (electronic supplementary material, figure S6). A clear significant correlation ($r = 0.55$; $p < 0.001$) was observed among all wild boar subpopulations in the southern and northern regions. However, no significant correlation was observed ($r = 0.03$; $p = 0.131$) for wild boar within the Fukushima evacuated zones. This result indicates that there is genetic mixture and similarity between the Fukushima evacuated area and northern prefecture's wild boar populations, while there is limited gene flow moving south.

## 4. Discussion

### (a) Genetic introgression from pigs, an invasive species, into wild boar

The data presented here reveal recent hybridization and genetic introgression from invasive pigs into wild boar. Hybrids and the extent of introgression from pigs were identified by maternally inherited markers and analysing multi-loci genotype data, which showed that 31 individuals, morphologically identified as wild boar, in Fukushima Prefecture had pig ancestry (figure 2). In a PCA analysis of wild boar and pigs (electronic supplementary material, figure S2), the

majority of the inferred hybrids were positioned between the wild boar cluster and pig cluster. The observed maternally inherited pig haplotype in two of the admixed individuals further support a scenario of hybridization.

A series of results showed that while a few individuals with pig ancestry were born into the Fukushima wild boar population, the proportion of pig genes in those individuals' genomes was low (average 8%, table 1). The proportion of hybrid wild boar in this dataset is 16%, which is higher than previously reported figures between 2 and 10% for introgression in European wild boar populations [15,42,43] and 10% for introgression in Ryukyu wild boar [44] using genetic markers. A higher frequency of hybrid wild boar was expected in this study because of the recent release of about 30 000 pigs into the wild following the Japanese Government ordered evacuations in 2011 [18]. However, despite a higher frequency of hybrids in our dataset, the estimated admixture or the average proportion of hybrids' genomes that originated from a pig population remained low (table 1) with all inferred hybrids containing an admixture of less than 50 and 30%, 5 and 7 years after the FDNPP accident, respectively (electronic supplementary material, table S1). Here, our results may be underestimating the full extent of introgression or hybrid occurrences because mtDNA data can only infer about female lineages (i.e. a cross between a female pig and a male wild boar) and our limiting sample size of pigs may not necessarily be representative of the complete gene pool of escaped pigs for STR. It may also be the case that our low pig ancestry results suggest a scenario that hybrids in this study were probably the offspring of one or more backcross generation of pure-wild boar, which would halve the number of introgressed pig alleles at every generation. While the selected STR markers consistently detected third backcross (BC2) generation hybrids (about 88%; see electronic supplementary material, table S2), it is feasible to assume a fourth or more backcross generation hybrid would likely be beyond the marker's detection ability.

The decrease of introgressed genes in the wild boars' genomes with the increase in the distance from the invasion source (figure 4) and with time is likely to be caused from the wild boar being most abundant in the evacuated zone [10] and the random mating tendencies of wild boar in this area (electronic supplementary material, figure S6), which in turn, increases the opportunity of backcross generations with wild boar and less chance of direct contact between remaining invasive pigs or other hybrids. Particularly, our result may be an indication of demic diffusion [25] in the wild, in which the invader gene pool is progressively diluted along the expansion due to recurrent admixture with the native species. This contrasts with reported hybridization in European wild boar populations where, despite new developments of intensive indoor pig farming, older hybridization events are being followed-up by interbreeding among hybrids, or recent escaped pigs, resulting in ongoing introgression on multiple occasions [42]. The difference in introgression observed in Europe versus our study may also be due to the rapid decline and total cessation of pig farming in Fukushima's evacuated zone because radiation dose rates exceed Japanese Government regulations. Therefore, the probability of follow-up hybridization events will remain low; direct contact between domestic pigs and wild boar is highly unlikely until pig farming practices are fully restored to pre-accident levels.

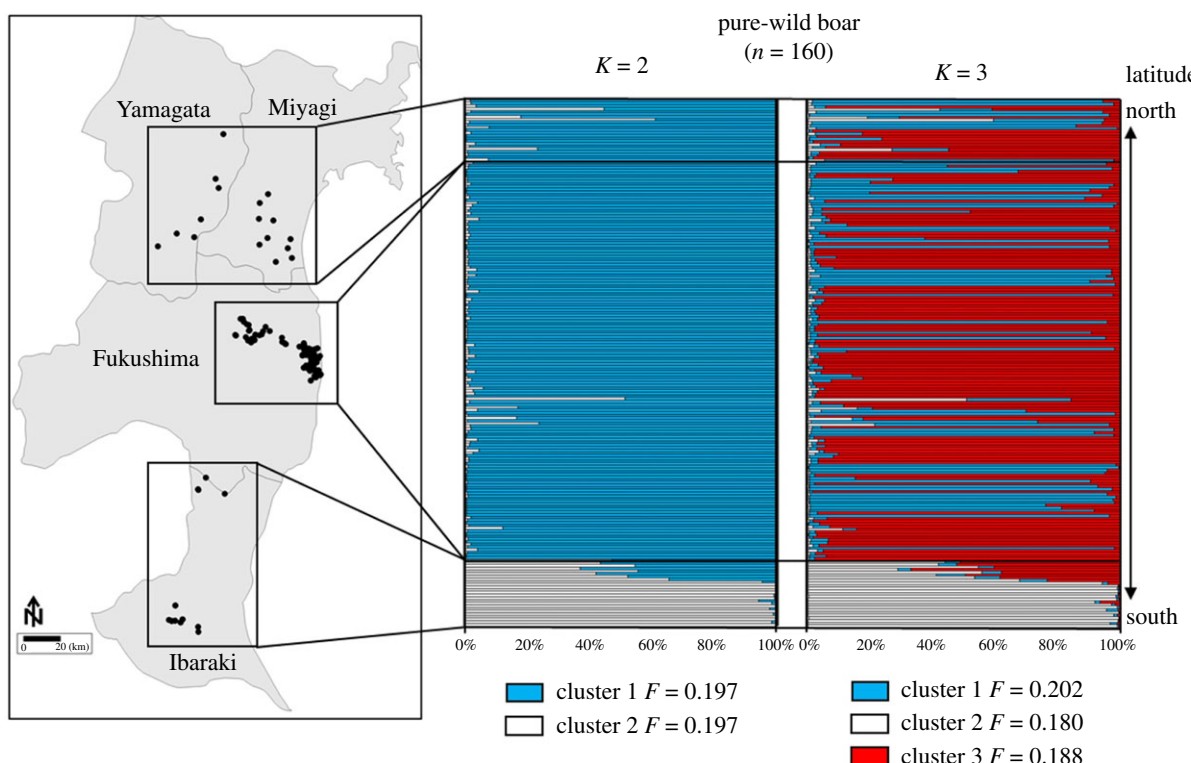

**Figure 5.** Distribution map of pure-wild boar populations and corresponding membership bar plots using STRUCTURE ($K = 2$ and $K = 3$) based on 24 loci. (Online version in colour.)

*Proc. R. Soc. B* **288**: 20210874

Despite such a sudden and large invasive force from pigs, our data likely suggest that most pigs failed to naturalize in the wild and survival of male pigs may have been higher. Here, we observed minimal selection of the pigs' invasion from the remaining invasive pig mtDNA haplotype in wild boar and the reduction of pig alleles, which is more pronounced in the STR loci because they are a mixture from both sexes. However, our results also detected only two individuals as hybrids by both the mtDNA and the STR markers. Previous studies have also pointed out inconsistencies in hybrid determination by mitochondrial and nuclear markers [15]. Given that our selected markers consistently detected third-generation backcrosses (see electronic supplementary material, table S2), we can speculate that female pigs and maternal offspring had difficulty surviving. If pig farms tended to raise more female pigs than male pigs, then this survival bias may explain the lack of concurrence in hybrid determination by mitochondrial and nuclear markers, and the seemingly unnatural distribution of $Q_1$ values in mitochondrial hybrids. Additionally, in the USA, male wild pigs tended to have higher survivability contributed to larger mass at birth compared to females [45]. Thus, it may be possible that male wild pigs or their paternal lineages in this area were more successful than female pigs.

## (b) Introgression source, dispersal prediction, and what this means for native wild boar

Our results suggest that abandoned livestock farms, inside the initial Japanese Government ordered evacuation zone (20 km radius from the FDNPP), were the source of the invasive pigs and the initial contact zone of introgression, with an indication of outward dispersal (figure 3). The highest number of hybrid wild boar was within this area and, on average, the hybrids in this area revealed the highest pig

ancestry (table 1). Thus, we can assume from this study that the hybrids have shown gradual range expansion from the contact zone while backcrossing on multiple occasions with wild boar.

When we compare our findings from the hybrid and pure-boar analyses to wild boar expansion trends reported in the literature, a probable prediction can be made of how hybrid lineages from pigs will disperse in the future. From the 1700s to 1970s, wild boar were thought to be regionally extinct in the northern mainland of Japan [1], which included our vast study area, presumably due to harsh winter climates and predatory populations. As climates warmed [45] and some predator species went extinct in Japan [46], wild boar expanded northward [1] filling new niches with little competition from conspecific species. Both our pure-wild boar genetic clustering (figure 5) and gene flow data among populations (electronic supplementary material, figure S6) inferred northern population expansion with limited gene flow moving south. We can predict, as climates warm, that wild boar and the offspring from hybrid lineages will follow a similar dispersal pattern north. Thus, if either the hybrid wild boar organelle inherited from invasive pigs or the introgressed pig alleles continue to persist with time, then we are more likely to observe these in northern wild boars' genomes at low frequencies.

The introgression of pig genes is suggested to have contributed to the rapid population growth rates of wild boar in Europe [23,47,48] and in the USA [49], and our evidence of introgression from pigs into the Japanese wild boars' gene pool should be a concern for conservation biologists, as increased abundances of wild boar may affect plant cover and influence food web dynamics in their environment [50,51]. However, our data also suggest that the introgression is most likely not ongoing in this area. Thus, an evolutionary shift due to the mixing of multiple lineages and predominant

expansion of intraspecies hybridization is not likely to occur in Japan as seen in Europe and the USA. If our highly supported prediction that the observed hybridization was sourced from the contact zone with reduced ancestry to pigs at greater distances, and if this is coupled with reduced pig farming in the area, then we can expect further dilution of remaining pig genes in the wild boar gene pool.

The greater abundance and higher densities of wild boar in the Fukushima evacuated area [10] coupled with our findings hybridization from invasive pigs establish a high-risk population to potential diseases [52,53]. As Japanese Government evacuation orders are lifted and human activities return to these landscapes, we can expect anthropogenic pressures to cause environmental stressors to wild boar and push expansion to less disruptive environments [4]. As this expansion occurs, the opportunities of disease transmission or contacting infected individuals may increase, which could be catastrophic for the high-density wild boar in this region. While there is limited information regarding the mechanisms of disease transmission [52], wild boar at similar population densities in Europe were mathematically modelled and estimated a 60–70% reduction in population density following an infectious outbreak [54]. Extensive introgression from pigs could alter wild boar immunology characteristics and resistance to these diseases [23] and although our investigation of introgression from pigs into the wild boar genome suggests the frequency of pig alleles has decreased at greater distances from the contact zone (table 1), the wild boar within the initial contact zone will likely continue to pass on introgressed pig alleles to the next generations. Therefore, the wild boar in this area should be periodically monitored using genetic markers, especially as pig farming communities are re-established and the chance of new hybridization events increase.

## 5. Conclusion

Here, using genetic markers, we demonstrated evidence of hybridization and introgression of invasive genes as a result of extrinsic environmental changes that reformed anthropogenic land use in Fukushima Prefecture in 2011. We found that there were likely successful hybridization events in the evacuation zone following a sudden and large biological invasion, that thereafter spread and diluted through time. However, a massive introgression was not observed in this area and we speculate two hypotheses for these patterns: (i) an abundant wild boar population caused increasing introgression of wild boar genes into the invasive pigs and a

decreasing introgression of invasive genes into the wild boar with the increase in the distance from the invasion source. (ii) The pig legacy passed on to the next-generation hybrids was dependent on the ability of pigs to naturalize in the wild area of Japan. In either case, if the invasion occurred in an environment that was not occupied already by high densities of wild boar, or if invaders had more advantageous survival traits for the wild, then maybe the invasive pigs would have had the same successful adaption as they had in other countries. We recommend that future studies assess the fitness of these hybrids and better characterize their ecological niche using range expansion models and their ecological interactions. Such studies could determine if demic diffusion occurred after range expansion, or if natural selection played a role. Both scenarios suggest that the introgressed genes will eventually disappear in this area.

Ethics. No animal was killed specifically for this research and all animals were legally culled by licensed hunters. The study was approved by Fukushima University's Institutional Animal Care and Use Committee. All experiments were performed in accordance with relevant guidelines and regulations.

Data accessibility. All data used in this study, supplemental tables and figures are provided in the electronic supplementary material available from the Dryad Digital Repository: https://doi.org/10.5061/dryad.qnk98sfgz [55].

Authors' contributions. D.A.: conceptualization, data curation, formal analysis, investigation, methodology, resources, software, validation, visualization, writing—original draft, writing—review and editing; Y.N.: data curation, formal analysis, investigation, visualization; H.I.: investigation, validation, writing—review and editing; K.O.: funding acquisition, investigation, resources, validation; T.G.H.: funding acquisition, investigation, resources, validation, writing—review and editing; R.T.: conceptualization, investigation, software; J.N.: resources, validation, writing—review and editing; H.B.T.: investigation, methodology, resources, validation; S.K.: conceptualization, data curation, formal analysis, funding acquisition, investigation, methodology, project administration, resources, software, supervision, validation, visualization, writing—review and editing.

All authors gave final approval for publication and agreed to be held accountable for the work performed therein.

Competing interests. We declare we have no competing interests.

Funding. Funding for this study was partially provided by the Nippon Life Insurance Foundation, and the Research Council of Norway through its Centres of Excellence funding scheme, project no. 223268/F50.

Acknowledgements. We are thankful to all prefectural hunters, and members of the Japanese Government-sponsored culling programme for their support in obtaining samples. We thank the Institute of Environmental Radioactivity at Fukushima University for their support in conducting this research. This work was supported by the support programme of FFPRI for researchers having family obligations.

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
