## [Peer Review File · Proceedings of the Royal Society B: Biological Sciences]

Review History

RSPB-2021-0383.R0 (Original submission)

Review form: Reviewer 1

Recommendation

Accept with minor revision (please list in comments)

Scientific importance: Is the manuscript an original and important contribution to its field?

Good

General interest: Is the paper of sufficient general interest?

Good

Quality of the paper: Is the overall quality of the paper suitable?

Good

Is the length of the paper justified?

Yes

Should the paper be seen by a specialist statistical reviewer?

No

Do you have any concerns about statistical analyses in this paper? If so, please specify them explicitly in your report.

No

It is a condition of publication that authors make their supporting data, code and materials available - either as supplementary material or hosted in an external repository. Please rate, if applicable, the supporting data on the following criteria.

Is it accessible?

Yes

Is it clear?

Yes

Is it adequate?

Yes

Do you have any ethical concerns with this paper?

No

Comments to the Author

Anderson et al. evaluated the level of introgression of domestic pigs into the wild boar population, following the Fukushima disaster, where thousands of domestic pigs from farms were released into nature. They described various population genetic variables in the area. They concluded that selection against the invasive gene pool would prevent long-term and massive introgression. While the manuscript was well-written overall, and the topic is relevant and of interest for publication, I have some concerns, which I will briefly describe, as follows:

My main concern is that selection was not properly tested in this manuscript, but was assumed to be the main explanation of not observing a massive introgression of domestic pigs into the wild boar population. The conclusion that "local adaption acted as resilience against a sudden and large biological invasion" might have been a bit overclaimed. An alternative explanation could be the result of different population dynamics, with one population (domestic pigs) in range expansion. In this last case, selection is not needed to explain the limited introgression observed into the local (wild boar) population, which could be explained by the neutral expectations of a population that is in range expansion. You might consider the paper of Quilodr an et al. 2020, *Frontiers in Ecology and Evolution*, for details about this alternative neutral explanation of your results.

Minor concerns:

- Lines 294-297; It would be better to show the coefficient of correlation (r) instead of the coefficient of determination (R^2).
- Line 329; "adaptation" instead of "adaption"
- Lines 384-387; would you provide more details about why an increase in the population growth-rate of native wild boars would be a conservation concern?

Review form: Reviewer 2

Recommendation

Reject - article is scientifically unsound

Scientific importance: Is the manuscript an original and important contribution to its field?
Acceptable

General interest: Is the paper of sufficient general interest?
Good

Quality of the paper: Is the overall quality of the paper suitable?
Poor

Is the length of the paper justified?
Yes

Should the paper be seen by a specialist statistical reviewer?
No

Do you have any concerns about statistical analyses in this paper? If so, please specify them explicitly in your report.
Yes

It is a condition of publication that authors make their supporting data, code and materials available - either as supplementary material or hosted in an external repository. Please rate, if applicable, the supporting data on the following criteria.

Is it accessible?
Yes

Is it clear?
Yes

Is it adequate?
Yes

Do you have any ethical concerns with this paper?
No

Comments to the Author

Dear Authors,

I find the topic covered by your work to be of wide interest, but i find many major flows that in my opinion doesn't make it suitable for publication in its present form.

First of all, the writing style is far from appropriate, with a number of unclear sentences throughout the ms, the use of abbreviations (SSRs) less common in this field compared to other ones (STRs), the use of different concepts (introgression vs hybridization) not always used in the most appropriate way, too numerous to be indicated (you will find them highlighted in yellow in the attached pdf). Some info presented in the Discussion should be better anticipated to the Introduction (e.g. number of escaped pigs, farms likely to have originated them, etc.). The fact that samples had been used in previous studies should be clearly stated in Matherials and Methods - Sampling section. Only 10 pigs (from a local market, but not necessarily representative of the gene pool of the escaped ones) have been analysed and this can strongly limit your ability to detect hybrids. The correct name for the software "Structure Harvester" is "Structure Harvester". A main weakness is that you define thresholds for identifying pure vs. hybrid boars and pigs before having identified the optimal number of clusters. Similarly, the analyses of geographical "populations" only based on the administrative borders lacks of significance and complicates the interpretation of data by the reader compared to the genetic clusters identified by Structure. Morevoer, it is not clear why in some analyses you considered a sub-optimal number of clusters (3 vs. 2). The difference between the average pig content at different distances from the

disaster zone was not tested for significance. Moreover, given the availability of samples also across time, I would expect data from different years to be compared in order to better trace the expected dilution effect. Overall, I think the manuscript could be highly improved in terms of data analyses, interpretation and style.

Decision letter (RSPB-2021-0383.R0)

25-Mar-2021

Dear Mr Anderson,

I am writing to inform you that your manuscript RSPB-2021-0383 entitled "Introgression dynamics from invasive pigs into wild boar following the March 2011 natural and anthropogenic disasters at Fukushima" has, in its current form, been rejected for publication in Proceedings B.

This action has been taken on the advice of referees, who have recommended that substantial revisions are necessary. With this in mind we would be happy to consider a resubmission, provided the comments of the referees are fully addressed. In particular, one of the reviewers raises very valid concerns about the statistical power of the study given your small sample size: see my further notes on this below. However please note that this is not a provisional acceptance, and I would recommend that you consider carefully if you can fully address the major concerns of Referee 2.

The resubmission will be treated as a new manuscript. However, we will approach the same reviewers. Please note that resubmissions must be submitted within six months of the date of this email. In exceptional circumstances, extensions may be possible if agreed with the Editorial Office. Manuscripts submitted after this date will be automatically rejected.

Yours sincerely,
Professor Loeske Kruuk
Editor
mailto: proceedingsb@royalsociety.org

Associate Editor

Board Member: 1

Comments to Author:

Thank you for your manuscript. We now have two reviews of this work. Both reviewers found the work and the system interesting but also had important critical comments both large and small. This will require extensive revision of the manuscript. Among the important points are about accounting for the year of sampling and age of the sampled individuals in their analyses, as well as the statistical power given the sample size of the analysis. I hope that you find the reviewers' comments useful and that the manuscript will improve greatly upon revision.

Editor (LKruuk)

Comments to Author(s):

I share the reviewers and AE's assessment of this being a very interesting study system and question, but also the concerns regarding the sample sizes and the analysis. The latter can be addressed with re-analysis, but regarding the sample sizes, please expand the sample size if at all possible. If not, you should explore the limitations of the sample size (you could get an indication of whether $n=10$ is limiting by comparing conclusions if you re-ran analyses with $n=5$ or $n=8$ pigs).

Reviewer(s)' Comments to Author:

Referee: 1

Comments to the Author(s)

Anderson et al. evaluated the level of introgression of domestic pigs into the wild boar population, following the Fukushima disaster, where thousands of domestic pigs from farms were released into nature. They described various population genetic variables in the area. They concluded that selection against the invasive gene pool would prevent long-term and massive introgression. While the manuscript was well-written overall, and the topic is relevant and of interest for publication, I have some concerns, which I will briefly describe, as follows:

My main concern is that selection was not properly tested in this manuscript, but was assumed to be the main explanation of not observing a massive introgression of domestic pigs into the wild boar population. The conclusion that "local adaptation acted as resilience against a sudden and large biological invasion" might have been a bit overclaimed. An alternative explanation could be the result of different population dynamics, with one population (domestic pigs) in range expansion. In this last case, selection is not needed to explain the limited introgression observed into the local (wild boar) population, which could be explained by the neutral expectations of a population that is in range expansion. You might consider the paper of Quilodr an et al. 2020, *Frontiers in Ecology and Evolution*, for details about this alternative neutral explanation of your results.

Minor concerns:

- Lines 294-297; It would be better to show the coefficient of correlation (r) instead of the coefficient of determination (R^2).
- Line 329; "adaptation" instead of "adaption"
- Lines 384-387; would you provide more details about why an increase in the population growth-rate of native wild boars would be a conservation concern?

Referee: 2

Comments to the Author(s)

Dear Authors,

I find the topic covered by your work to be of wide interest, but i find many major flaws that in my opinion doesn't make it suitable for publication in its present form.

First of all, the writing style is far from appropriate, with a number of unclear sentences throughout the ms, the use of abbreviations (SSRs) less common in this field compared to other ones (STRs), the use of different concepts (introgression vs hybridization) not always used in the most appropriate way, too numerous to be indicated (you will find them highlighted in yellow in

the attached pdf). Some info presented in the Discussion should be better anticipated to the Introduction (e.g. number of escaped pigs, farms likely to have originated them, etc.). The fact that samples had been used in previous studies should be clearly stated in Materials and Methods - Sampling section. Only 10 pigs (from a local market, but not necessarily representative of the gene pool of the escaped ones) have been analysed and this can strongly limit your ability to detect hybrids. The correct name for the software "Structure Harvester" is "Structure Harvester". A main weakness is that you define thresholds for identifying pure vs. hybrid boars and pigs before having identified the optimal number of clusters. Similarly, the analyses of geographical "populations" only based on the administrative borders lacks of significance and complicates the interpretation of data by the reader compared to the genetic clusters identified by Structure. Moreover, it is not clear why in some analyses you considered a sub-optimal number of clusters (3 vs. 2). The difference between the average pig content at different distances from the disaster zone was not tested for significance. Moreover, given the availability of samples also across time, I would expect data from different years to be compared in order to better trace the expected dilution effect. Overall, I think the manuscript could be highly improved in terms of data analyses, interpretation and style.

Author's Response to Decision Letter for (RSPB-2021-0383.R0)

See Appendix A.

RSPB-2021-0874.R0

Review form: Reviewer 2

Recommendation

Major revision is needed (please make suggestions in comments)

Scientific importance: Is the manuscript an original and important contribution to its field?

Acceptable

General interest: Is the paper of sufficient general interest?

Good

Quality of the paper: Is the overall quality of the paper suitable?

Marginal

Is the length of the paper justified?

Yes

Should the paper be seen by a specialist statistical reviewer?

No

Do you have any concerns about statistical analyses in this paper? If so, please specify them explicitly in your report.

Yes

It is a condition of publication that authors make their supporting data, code and materials available - either as supplementary material or hosted in an external repository. Please rate, if applicable, the supporting data on the following criteria.

Is it accessible?

Yes

Is it clear?

Yes

Is it adequate?

Yes

Do you have any ethical concerns with this paper?

Yes

Comments to the Author

The authors tried hard to improve their ms, partly reaching this goal. Still, there are several passages to be further improved.

L. 29-30: highly speculative here

L. 37: "overall loss"

L. 52: repetition

L. 79: i don't think that in the evacuation area you can consider multiple "populations"

L. 80-81: please report any available estimate, in order to be comparable with the number of escaped pigs

L. 96-97: unclear, please rephrase

L. 138: delete "studies"

L. 178: please motivate why you used correlated allele frequencies even dealing with a wild population and a domestic one

L. 197: mtDNA inheritance can be representative of either hybridization AND introgression; the same applies to STR admixed genotypes. I'd keep both the correct definitions provided, but without referencing to these misleading examples

L. 199-201: again, a boar with pig mtDNA but 100% STR assignment to wild boar pop is an example of introgression

L. 219-221: please consider whether it is worth using the samples collected before 2011 separately in order to evaluate possible genetic drift

L. 247-248: please postpone after you have defined the two reference populations in the following sentence

L. 268-270: what about the other 21 hybrids? Why didn't you include them in this comparison?

You have these data available, please use it

L. 273: typo

L. 278-279: Please explain better (as you did in the answer to reviewers)

L. 286-289: Please, highlight where pre-2011 individuals fall among these clusters

L. 319: "pig ancestry" instead of "admixture"

L. 328-329: as you stated in the response letter, bred pigs probably show limited diversity compared to wild boars, please be consistent

L. 331: "favoured" instead of "preferred"

L. 333: "under" instead of "during"

L. 333-339: all this part is unclear and/or the assumptions are not properly supported. Please, carefully review it and make it coherent with the following section (is the observed pattern of decrease of pig ancestry compatible with selection or with neutral dilution under random mating? Based on the current analyses, you have no chance to document selection unless you formally test for it

L. 366: "pig ancestry" instead of "admixture to the pig population"

L. 377: "genetic mixture" --– do you mean "gene flow"?

- L. 395: "genetically similar populations"  not clear
 L. 417: "hybridization" instead of "introgression"
 L. 420-423: all unclear
 L. 423: i guess 30.000 is hardly definable as "limited"

Review form: Reviewer 3

Recommendation

Major revision is needed (please make suggestions in comments)

Scientific importance: Is the manuscript an original and important contribution to its field?

Excellent

General interest: Is the paper of sufficient general interest?

Excellent

Quality of the paper: Is the overall quality of the paper suitable?

Good

Is the length of the paper justified?

Yes

Should the paper be seen by a specialist statistical reviewer?

No

Do you have any concerns about statistical analyses in this paper? If so, please specify them explicitly in your report.

Yes

It is a condition of publication that authors make their supporting data, code and materials available - either as supplementary material or hosted in an external repository. Please rate, if applicable, the supporting data on the following criteria.

Is it accessible?

Yes

Is it clear?

Yes

Is it adequate?

Yes

Do you have any ethical concerns with this paper?

No

Comments to the Author

The paper 'Introgression dynamics from invasive pigs into wild boar following the March 2011 natural and anthropogenic disasters at Fukushima' asks about the hybridization between wild boar and domestic pigs that might have occurred since the Fukushima melt down in 2011. I like this paper. It's clear and well written and reasoned, and asks a really interesting question. I have a few points of clarification on some of the analyses, which, I hope are straightforward to address.

I would really like to see this paper address the power of the 24 STR markers that are used in the nuclear DNA analysis. Are these markers highly diverged, diagnostic or ancestry informative between wild boar and pigs? Or are they just polygenic and amplifying in both? I would describe the allele frequencies for each of these markers in pure species individuals. What really caught my eye is that 1) Table 1 appears to only have two individuals that were detected as hybrids by both the mtDNA and the STR markers and 2) the few 'suggested hybrids' (i.e. those indicated as hybrids in some runs of STRUCTURE, but not all) have substantially lower Q scores than the hybrids consistently detected by STRUCTURE. I can imagine a couple of explanations for these patterns.

The first is that the 24 markers are not powerful enough to detect later generation backcrosses consistently. Based on a quick search, I think that these animals have a generation time of 2- 3 years, so I estimate that around 3rd generation backcrosses could be common in this population. 3rd generation backcrosses have an average Q score of ~0.06. Do these 24 markers consistently detect those individuals? Could there have been even more generations of backcrossing elapsed since the Fukushima disasters? I really have no idea, but this would be worth addressing. I think that the best way to do this would be to include credible intervals on the STRUCTURE plot (Figure 2), and in Table S1 beside the Q scores for all hybrids. Boecklen and Howard (1997) have a nice deterministic model describing how many markers are needed to detect backcrossing and Vähä and Primmer (2006) also describe this problem and use simulations to make suggestions. The manuscript hints a bit at this general problem on lines 319-321, but I do think it is addressable to some extent.

The other explanation that I can imagine, that would be really interesting to discuss (but I suspect speculative!) is that much of the recent hybridization, aka the hybridization in the last 10 years, is usually between male pigs and female boars, so there is no mtDNA introgression in the nuclear hybrids detected. This wouldn't explain why the mt-hybrids weren't detected using STRs, except, as the authors noted, that could be related to very old hybridization, or, as above, could have only happened at the initial hybridization event 10 years ago, and thus is beyond the detection ability of the 24 STRs. I'm not sure, but it could be cool to explore this idea. The manuscript notes 'random mating tendencies' (line 339), but this is just based on relatedness, isn't it? Is there assortative mating for species? If male pigs are more successful than female pigs, or if a huge proportion of domestic pigs died in 2011 (I can imagine they didn't fare well without being fed), perhaps the piglets of female boars who had pig sires did better than the piglets of female pigs x male boars (which would be carrying the pig mtDNA, as well as substantial nuclear DNA).

In either case, I think that the lack of concurrence between the nuclear and mt hybrids ought to be addressed. I think it's either a lack of power from the STRs, or an interesting biological mechanism that could only be suggested once a lack of power was ruled out.

Boecklen, W. J. and D. J. Howard. 1997. Genetic analysis of hybrid zones: numbers of markers and power of resolution. *Ecology* 78:2611-2616.

Vähä, J.-P. and C. R. Primmer. 2006. Efficiency of model-based Bayesian methods for detecting hybrid individuals under different hybridization scenarios and with different numbers of loci. *Molecular Ecology* 15:63-72.

Decision letter (RSPB-2021-0874.R0)

28-May-2021

Dear Mr Anderson:

Your manuscript has now been peer reviewed and the reviews have been assessed by an Associate Editor. The reviewers' comments (not including confidential comments to the Editor) and the comments from the Associate Editor are included at the end of this email for your reference. As you will see, the reviewers have raised some concerns with your manuscript and we would like to invite you to revise your manuscript to address them.

Research ethics:

Use of animals and field studies:

It is a condition of publication that you make available the data and research materials supporting the results in the article (<https://royalsociety.org/journals/authors/author-guidelines/#data>). Datasets should be deposited in an appropriate publicly available repository and details of the associated accession number, link or DOI to the datasets must be included in the Data Accessibility section of the article (<https://royalsociety.org/journals/ethics-policies/data-sharing-mining/>). Reference(s) to datasets should also be included in the reference list of the article with DOIs (where available).

Please submit a copy of your revised paper within three weeks. If we do not hear from you within this time your manuscript will be rejected. If you are unable to meet this deadline please let us know as soon as possible, as we may be able to grant a short extension.

Best wishes,
Professor Loeske Kruuk
mailto:proceedingsb@royalsociety.org

Associate Editor Board Member: 1

Comments to Author:

The reviewers both feel that this paper is improving and is highly likely to make an important contribution to this field. There are a number of small points and two large conceptual ones remaining. The two big ones are: 1) address why so few of their hybrids were detected by both markers and 2) be more transparent about the power of their STRs, by looking at their population specific allele frequencies and reporting CIs. The reviewers also identified a number of more minor issues.

Reviewer(s)' Comments to Author:

Referee: 2

Comments to the Author(s).

The authors tried hard to improve their ms, partly reaching this goal. Still, there are several passages to be further improved.

L. 29-30: highly speculative here

L. 37: "overall loss"

L. 52: repetition

L. 79: i don't think that in the evacuation area you can consider multiple "populations"

L. 80-81: please report any available estimate, in order to be comparable with the number of escaped pigs

L. 96-97: unclear, please rephrase

- L. 138: delete "studies"
- L. 178: please motivate why you used correlated allele frequencies even dealing with a wild population and a domestic one
- L. 197: mtDNA inheritance can be representative of either hybridization AND introgression; the same applies to STR admixed genotypes. I'd keep both the correct definitions provided, but without referencing to these misleading examples
- L. 199-201: again, a boar with pig mtDNA but 100% STR assignment to wild boar pop is an example of introgression
- L. 219-221: please consider whether it is worth using the samples collected before 2011 separately in order to evaluate possible genetic drift
- L. 247-248: please postpone after you have defined the two reference populations in the following sentence
- L. 268-270: what about the other 21 hybrids? Why didn't you include them in this comparison? You have these data available, please use it
- L. 273: typo
- L. 278-279: Please explain better (as you did in the answer to reviewers)
- L. 286-289: Please, highlight where pre-2011 individuals fall among these clusters
- L. 319: "pig ancestry" instead of "admixture"
- L. 328-329: as you stated in the response letter, bred pigs probably show limited diversity compared to wild boars, please be consistent
- L. 331: "favoured" instead of "preferred"
- L. 333: "under" instead of "during"
- L. 333-339: all this part is unclear and/or the assumptions are not properly supported. Please, carefully review it and make it coherent with the following section (is the observed pattern of decrease of pig ancestry compatible with selection or with neutral dilution under random mating? Based on the current analyses, you have no chance to document selection unless you formally test for it
- L. 366: "pig ancestry" instead of "admixture to the pig population"
- L. 377: "genetic mixture"  do you mean "gene flow"?
- L. 395: "genetically similar populations"  not clear
- L. 417: "hybridization" instead of "introgression"
- L. 420-423: all unclear
- L. 423: i guess 30.000 is hardly definable as "limited"

Referee: 3

Comments to the Author(s).

The paper 'Introgression dynamics from invasive pigs into wild boar following the March 2011 natural and anthropogenic disasters at Fukushima' asks about the hybridization between wild boar and domestic pigs that might have occurred since the Fukushima melt down in 2011. I like this paper. It's clear and well written and reasoned, and asks a really interesting question. I have a few points of clarification on some of the analyses, which, I hope are straightforward to address.

I would really like to see this paper address the power of the 24 STR markers that are used in the nuclear DNA analysis. Are these markers highly diverged, diagnostic or ancestry informative between wild boar and pigs? Or are they just polygenic and amplifying in both? I would describe the allele frequencies for each of these markers in pure species individuals. What really caught my eye is that 1) Table 1 appears to only have two individuals that were detected as hybrids by both the mtDNA and the STR markers and 2) the few 'suggested hybrids' (i.e. those indicated as hybrids in some runs of STRUCTURE, but not all) have substantially lower Q scores than the hybrids consistently detected by STRUCTURE. I can imagine a couple of explanations for these patterns.

The first is that the 24 markers are not powerful enough to detect later generation backcrosses consistently. Based on a quick search, I think that these animals have a generation time of 2- 3 years, so I estimate that around 3rd generation backcrosses could be common in this population.

3rd generation backcrosses have an average Q score of ~ 0.06 . Do these 24 markers consistently detect those individuals? Could there have been even more generations of backcrossing elapsed since the Fukushima disasters? I really have no idea, but this would be worth addressing. I think that the best way to do this would be to include credible intervals on the STRUCTURE plot (Figure 2), and in Table S1 beside the Q scores for all hybrids. Boecklen and Howard (1997) have a nice deterministic model describing how many markers are needed to detect backcrossing and Vähä and Primmer (2006) also describe this problem and use simulations to make suggestions. The manuscript hints a bit at this general problem on lines 319-321, but I do think it is addressable to some extent.

The other explanation that I can imagine, that would be really interesting to discuss (but I suspect speculative!) is that much of the recent hybridization, aka the hybridization in the last 10 years, is usually between male pigs and female boars, so there is no mtDNA introgression in the nuclear hybrids detected. This wouldn't explain why the mt-hybrids weren't detected using STRs, except, as the authors noted, that could be related to very old hybridization, or, as above, could have only happened at the initial hybridization event 10 years ago, and thus is beyond the detection ability of the 24 STRs. I'm not sure, but it could be cool to explore this idea. The manuscript notes 'random mating tendencies' (line 339), but this is just based on relatedness, isn't it? Is there assortative mating for species? If male pigs are more successful than female pigs, or if a huge proportion of domestic pigs died in 2011 (I can imagine they didn't fare well without being fed), perhaps the piglets of female boars who had pig sires did better than the piglets of female pigs x male boars (which would be carrying the pig mtDNA, as well as substantial nuclear DNA).

In either case, I think that the lack of concurrence between the nuclear and mt hybrids ought to be addressed. I think it's either a lack of power from the STRs, or an interesting biological mechanism that could only be suggested once a lack of power was ruled out.

Boecklen, W. J. and D. J. Howard. 1997. Genetic analysis of hybrid zones: numbers of markers and power of resolution. *Ecology* 78:2611-2616.

Vähä, J.-P. and C. R. Primmer. 2006. Efficiency of model-based Bayesian methods for detecting hybrid individuals under different hybridization scenarios and with different numbers of loci. *Molecular Ecology* 15:63-72.

Author's Response to Decision Letter for (RSPB-2021-0874.R0)

See Appendix B.

Decision letter (RSPB-2021-0874.R1)

07-Jun-2021

Dear Mr Anderson

I am pleased to inform you that your manuscript entitled "Introgression dynamics from invasive pigs into wild boar following the March 2011 natural and anthropogenic disasters at Fukushima" has been accepted for publication in *Proceedings B*.

You can expect to receive a proof of your article from our Production office in due course, please check your spam filter if you do not receive it. PLEASE NOTE: you will be given the exact page

length of your paper which may be different from the estimation from Editorial and you may be asked to reduce your paper if it goes over the 10 page limit.

Data Accessibility section

Open Access

Paper charges

Yours sincerely,
Professor Loeske Kruuk
Editor, Proceedings B
<mailto:proceedingsb@royalsociety.org>

Associate Editor:

Comments to Author:

Thank you for the careful attention to the comments in the previous round. I am confident this manuscript will make an impact on this area of the literature and be useful for researchers investigating introgression dynamics in many other systems.

Appendix A

Response to referees' letter

Journal Name: Proceedings of the Royal Society B

Journal Code: RSPB

MS Reference Number: RSPB-2021-0383

Title: Introgression dynamics from invasive pigs into wild boar following the March 2011 natural and anthropogenic disasters at Fukushima

Dear Professor Loeske Kruuk,

First, we would like to thank the Associate Editor, the reviewers, and you for the letter sent on 25th March 2021 regarding our manuscript (**RSPB-2021-0383**) entitled “Introgression dynamics from invasive pigs into wild boar following the March 2011 natural and anthropogenic disasters at Fukushima” by Anderson, Donovan; Negishi, Yuki; Ishiniwa, Hiroko; Okuda, Kei; Hinton, Thomas G.; Toma, Rio; Nagata, Junco; Tamate, Hidetosh B.; Kaneko, Shingo. We have thoroughly revised our manuscript in response to the suggestions of the Associate Editor and the reviewers. The details of improvements, revisions, and how we responded to their individual comments are presented in the remainder of this letter. In addition, we hope we have adequately addressed the major and valid concerns about the statistical power of our study with regards to the sample size, and the major concerns from Referee 2.

We hope that you will find the revised version suitable for publication in the Proceedings of the Royal Society B.

Please note that Line numbers will refer to attached file ‘manuscript marked with changes.’

Editor (LKruuk),

Again, thank you for your comments and review of our manuscript. Below, we have investigated the limitations of our sample size as suggested. Your shared concerns are addressed in responses to the AE's and reviewer's assessments below.

Editor comments to Author(s):

I share the reviewers and AE's assessment of this being a very interesting study system and question, but also the concerns regarding the sample sizes and the analysis. The latter can be addressed with re-analysis, but regarding the sample sizes, please expand the sample size if at all possible. If not, you should explore the limitations of the sample size (you could get an indication of whether $n=10$ is limiting by comparing conclusions if you re-ran analyses with $n=5$ or $n=8$ pigs).

Here, we have investigated the limitations of our sample size by re-running the STRUCTURE analysis with $n=6, 7, 8,$ and 9 pigs and then comparing conclusions with $n=10$ pigs. Each of the varying sample size scenarios were performed 10 times. We provide the results below from the three best STRUCTURE results with the highest LnPD. In addition, the pigs used in each analysis were chosen at random. All re-run analyses were performed using a Bayesian clustering algorithm available in STRUCTURE v. 2.3.4 (As described in the manuscript (Lines 194-197)).

“STRUCTURE was run using the settings of the admixture model with correlated allele frequencies; all parameters were set to default with the LOCPRIOR model implemented. To estimate the number of clusters (K), 10 independent runs with $K = 1-10$ were performed using 10^6 Markov Chain Monte Carlo (MCMC) iterations following a burn-in of 10^5 iterations.”

While we initially shared the concerns that only 10 pigs would limit the ability to detect hybrids, our comparative results reduced number of samples reaffirm our results. We also found indications that a sample size of n=6 pigs would be limiting. At n=6, the number of hybrids detected and proportion of the genome originating from pigs were too high. Thus, we can safely assume that a sample set of 6 pigs is probably too small. However, results with n=7-9 compare well our findings. Thus, we are confident that a sample set of 8 or more pigs is reasonable. At n=8, and n=9 the pig samples were able to detect the same number of hybrid individuals as our n = 10 analysis (see Response Table 1 below). It is likely that adding more pigs will not largely influence or change allele frequencies and the shared alleles between pig and boar still remain limited in this area. This is likely the case because the pig populations are not under random mating, but highly selected or controlled populations in the agriculture setting. At n=8 we were able to detect the same hybrids, which also infers our markers for hybrid detection were adequate for these populations in our study area (see Anderson D, Negishi Y, Toma R, Nagata J, Tamate H. 2020 Robust microsatellite markers for hybrid analysis between domesticated pigs and wild boar. *Genet. Resour.* **1**, 29–41. (doi:10.46265/genresj.BNHB8715).

However, we understand that the Q_1 values or number of hybrids detected, were likely an underestimate due to our limited pig sample size, and now we address this in the manuscript for readers in the Discussion (Line 379-383).

Added text (Discussion, Line 379-383);

“Here, our results may be underestimating the full extent of introgression or hybrid occurrences because mtDNA data can only infer about female lineages (i.e., a cross between a female pig and a male wild boar) and our limiting sample size of pigs may not necessarily be representative of the complete gene pool of escaped pigs for STR.”

Also, we also decided to define better thresholds and definitions for hybrids based on all our 10 independent runs of STRUCTURE at n=10 in the manuscript. This reduces the changing data by each trial run, especially when admixed individuals have a low estimated proportion of their genome related to pig (~1%). Our results/conclusions did not change with this amendment, but it is now more clear for readers.

Added text (Materials and methods) Line 205-213; “We assigned each individual to the inferred clusters, using a conservative threshold $Q_1 \geq 0.99$ for the assignment of individuals genomes to the pig cluster, or, $Q_1 < 0.01$ to the wild boar cluster. Admixed individuals were jointly assigned to the two clusters and were considered to have possible hybrid ancestry or were the offspring from ancestors that were a cross between wild boar and pig. Admixed individuals were considered hybrids only if the neither of the thresholds were true in all independent runs. If an individual admixed in more than 50% of the runs, but not all, then the individual was considered as a “suggested hybrid.”

Table Response 1: The STRUCTURE results using n=6,7,8,9 and 10 pig samples. We ran STRUCTURE 10 times for each n and present the 3 best trials for each n based on the highest estimated Ln. The highlighted yellow cells would indicate an admixed individual (boar x pig) or probable hybrid. Grey cells indicate hybrids that admixed in 100% of the independent runs. Light blue cells indicate “suggested hybrids” or individuals that admixed in >50% of the trials. In this case, this matched our manuscript data and the independent runs of n=10 performed therein. Thus, we are confident that our sample size is not a limiting factor. However, the changes in admixed individuals with very genomes originating from pig (~0.005-0.03) with each trial run required us to amend the manuscript to more conservative thresholds and definition of hybrid vs “suggested hybrid.” (Please see above, Added text (Materials and methods) Line 205-213).

Sample ID	Number of runs detected Hybrid	n=10	n=10	n=10	n=9	n=9	n=9	n=8	n=8	n=8	n=7	n=7	n=7	n=6	n=6	n=6
D012	8	0.018	0.004	0.003	0.003	0.019	0.004	0.008	0.010	0.004	0.028	0.017	0.021	0.001	0.023	0.022
D032	10	0.027	0.008	0.007	0.008	0.031	0.009	0.017	0.021	0.010	0.040	0.024	0.031	0.003	0.036	0.037
D046	15	0.129	0.104	0.112	0.112	0.132	0.105	0.115	0.128	0.115	0.123	0.086	0.106	0.035	0.112	0.114
D048	15	0.246	0.238	0.246	0.254	0.252	0.250	0.257	0.258	0.259	0.259	0.259	0.258	0.265	0.266	0.267
D060	6	0.013	0.003	0.002	0.002	0.011	0.002	0.005	0.006	0.002	0.016	0.009	0.012	0.001	0.013	0.013
D065	9	0.026	0.009	0.008	0.007	0.025	0.008	0.012	0.016	0.008	0.031	0.020	0.025	0.003	0.026	0.026
D090	15	0.165	0.153	0.159	0.167	0.170	0.163	0.171	0.173	0.171	0.182	0.179	0.180	0.180	0.187	0.188
D091	15	0.066	0.036	0.038	0.041	0.069	0.039	0.047	0.058	0.045	0.083	0.060	0.073	0.031	0.077	0.078
D127	15	0.314	0.306	0.316	0.325	0.320	0.320	0.328	0.330	0.331	0.335	0.336	0.335	0.345	0.345	0.348
D132	15	0.332	0.324	0.334	0.342	0.337	0.337	0.344	0.345	0.346	0.348	0.350	0.348	0.356	0.355	0.357
K024	14	0.031	0.011	0.010	0.010	0.033	0.011	0.017	0.022	0.011	0.048	0.030	0.038	0.006	0.042	0.042
K036	2	0.002	0.000	0.000	0.000	0.003	0.001	0.004	0.003	0.001	0.005	0.003	0.004	0.000	0.012	0.014
K038	2	0.005	0.001	0.001	0.001	0.007	0.001	0.007	0.006	0.001	0.014	0.009	0.012	0.001	0.023	0.026
K064	5	0.011	0.003	0.003	0.002	0.011	0.003	0.005	0.006	0.002	0.013	0.008	0.010	0.001	0.011	0.010
K122	10	0.031	0.009	0.008	0.009	0.035	0.009	0.017	0.022	0.010	0.047	0.029	0.037	0.003	0.039	0.039
K124	15	0.243	0.238	0.246	0.252	0.247	0.248	0.253	0.254	0.254	0.256	0.258	0.257	0.264	0.263	0.265
K127	15	0.379	0.371	0.383	0.393	0.385	0.387	0.394	0.395	0.396	0.400	0.403	0.400	0.412	0.410	0.412
K132	15	0.502	0.495	0.510	0.514	0.502	0.507	0.509	0.511	0.514	0.518	0.524	0.519	0.534	0.526	0.529
K169	15	0.058	0.030	0.030	0.019	0.047	0.019	0.026	0.033	0.020	0.053	0.034	0.043	0.008	0.046	0.046
K173	1	0.005	0.001	0.001	0.001	0.006	0.001	0.002	0.003	0.001	0.011	0.007	0.009	0.000	0.008	0.008

In addition, to further support our sample size, our PCoA analysis of wild boar and pigs clearly distinguished the pig samples (**Fig. S2** or below) and most assumed hybrids were positioned separately from the wild boar cluster, trailing off in the direction of the pig cluster. These results would not be influenced by the sample size of pigs.

To the Associate Editor,

Thank you very much for your coordinating roles as an associate editor. We truly appreciate the time you and two reviewers invested in reviewing our manuscript so carefully. We have responded to all of the individual editorial and substantive comments by the two reviewers and have extensively revised our manuscript accordingly. All the comments were useful and greatly improved the manuscript.

AE's comments to Author(s):

Thank you for your manuscript. We now have two reviews of this work. Both reviewers found the work and the system interesting but also had important critical comments both large and small. This will require extensive revision of the manuscript. Among the important points are about accounting for the year of sampling and age of the sampled individuals in their analyses, as well as the statistical power given the sample size of the analysis. I hope that you find the reviewers' comments useful and that the manuscript will improve greatly upon revision.

Response to: "as well as the statistical power given the sample size of the analysis";

In regards to the statistical power given the sample size of our analysis, please see our response to the editor above. In brief, while we initially shared the concerns of our limited sample size, and its ability to detect hybrids, our re-analyses using $n=6,7,8$, and 9 pig samples compared to $n=10$, as suggested by the Editor revealed that our sample size is not a limiting factor. After this analysis we are confident in our data analysis with 10 samples, as results were consistent after $n=7$ (see Response Table 1 above). This indicates that our markers are able to successfully detect the rare shared alleles between the released pigs and wild boar populations in Fukushima, which would be the indication of hybridization. If the pigs were under a wild and random-mating scenario then likely a greater number of individuals would be needed to detect hybrids. In this case, the pigs were in a controlled and highly selective setting, thus increasing the number of individuals will most likely not change allele frequencies and shared/introgressed alleles between pig and boar still remain limited. We found that a few individuals admixed in some independent runs, but not in others. Thus, we have made new thresholds and definitions for hybrid vs "suggested hybrid." This is also described earlier in the letter.

However, we do understand these concerns stated by your comment and that this is still a possible limitation to the study. So, now we present this limitation in the manuscript that the number of hybrids detected is likely an underestimate due to our sample size of pigs. We address this in the manuscript for readers in the **Discussion** (Line 379-383).

Added text (**Discussion** Line 379-383);

“Here, our results may be underestimating the full extent of introgression or hybrid occurrences because mtDNA data can only infer about female lineages (i.e., a cross between a female pig and a male wild boar) and our limiting sample size of pigs may not necessarily be representative of the complete gene pool of escaped pigs for STR. It may also be the case...”

Response to: “Among the important points are about accounting for the year of sampling and age of the sampled individuals in their analyses”;

There was a shared interest/concern to account for the year sampling and age of the sampled individuals in our analyses. We would like to mention that the estimated age data coupled with genetic data, for a large mammal in the wild, is unique and typically not available in genetic studies. So, we thought it would add to the manuscript. So, prior to initial submission we performed the statistical analyses, however we felt that while these analyzes may provide new perspectives, they are vulnerable to the limited number of hybrids detected by STRs and do not affect the main conclusions of this paper. In other words, we felt that 1., since sample locations and collected year is limited by the hunter's convenience and not of biological reasons (number of populations is not change) and 2., the age of boar would not make significant difference due to the random mating of populations (Fig S6) and quick maturity (less than one year) and birth periods for Japanese wild boar (see Kodera Y, Takeda T, Tomaru S, Sugita S. 2012 The estimation of birth periods in wild boar by detailed aging. *Mamm. Sci.* **52**, 185–191. (<https://doi.org/10.11238/mammalianscience.52.185>)). Thus, we felt the outcome of these analyses were not appropriate for presentation.

In hope to improve the manuscript in light of these shared concerns/interest we now:

- 1.) Subtracted the estimated age of the boar from the sampling month-year to approximate the period of birth, henceforth referred to as “birth year” of each hybrid boar. In other words, if a hybrid boar was sampled in July 2016 and was estimated to be 52 weeks old (~1 year) then the approximate birth year was July-2015. This new approximate birth year may help account for the lack of biological significance from year sampled. Using this birth year and pig content, we were able to provide some evidence of the dilution effect, as also mentioned by the second reviewer's comment. However, there remains a poor correlation between admixture and birth year ($r = 0.05$), likely due to the small number of hybrids detected. The other possible reasoning for this is because, while our markers are able to detect hybrids, the released pigs may have been the progeny of prior hybrids due to the artificial selection/mixing of genes from boar to pigs by the farming industry as evidenced of Asian boar haplotypes in our pig samples (Anderson D, Negishi Y, Toma R, Nagata J, Tamate H. 2020 Robust microsatellite markers for hybrid analysis between domesticated pigs and wild boar. *Genet. Resour.* **1**, 29–41. (doi:10.46265/genresj.BNHB8715)). In this case, pig alleles are hidden in shared alleles or lost by backcross generations, and our markers cannot detect such hybrids.
- 2.) We now provide the age of hybrid boar and “birth year” in amended **Table S1**, and these data can be used for future reference.

Using the new birth year, we have now added the text to manuscript:

Added text Results Line 309-312: “Using the approximated birth year of hybrid wild boars identified by STR analysis, five hybrids were approximated to be birthed in 2016 that ranged from 0.03 to 0.51 (average = 0.27); and five hybrids in 2018 that ranged from 0.04 to 0.33 (average = 0.21) (Table S1, electronic supplementary material).”

To Reviewer 1,

Thank you for your helpful comments. We appreciate your careful review of our manuscript. We have revised our manuscript thoroughly by attending to all of your individual comments. Primarily, we investigated your suggestion to the alternative explanation as to why there was not an observation of a massive introgression in our results.

Major Concerns:

My main concern is that selection was not properly tested in this manuscript, but was assumed to be the main explanation of not observing a massive introgression of domestic pigs into the wild boar population. The conclusion that “local adaption acted as resilience against a sudden and large biological invasion” might have been a bit overclaimed. An alternative explanation could be the result of different population dynamics, with one population (domestic pigs) in range expansion. In this last case, selection is not needed to explain the limited introgression observed into the local (wild boar) population, which could be explained by the neutral expectations of a population that is in range expansion. You might consider the paper of Quilodrán et al. 2020, *Frontiers in Ecology and Evolution*, for details about this alternative neutral explanation of your results.

We first would like to again thank the reviewer for their comments. Here, we agree with the reviewer that the conclusion “local adaption acted as resilience against a sudden and large biological invasion” was overclaimed. In light of this, we have amended the manuscript accordingly to adjust this (see below edits). In addition, we have investigated the reviewer’s alternative explanation about range expansion for the explanation of our results in this area. Following review of Quilodrán et al. 2020, in addition to other range expansion articles, different population dynamics, with one population (pigs) in range expansion could explain the limited introgression or hybridization in this area. The explanation is now discussed in the manuscript as a possible reason for our results. In addition to this, we have rewritten our conclusion.

Revisions made regarding this comment:

Abstract

Added text (Line 26-29): “Concurrently, we show how invasive pigs failed to naturalize in the same landscapes and, despite evidence of successful hybridization between pigs and native wild boar in this area, in future offspring, **selection native gene pool or range expansion dynamics inhibit** long-term introgression.”

Introduction

Added text (Line 76-79): “The interactions and genetic mixing from intraspecies hybridization can be used to provide key insight on the naturalization, **range expansion dynamics**, and natural selection phenomena of influenced species in these landscapes.”

Amended text (Line 98-101): “**Thus, investigating the influence of intraspecies hybridization and introgression here, a natural population influenced by a recent large invasion, could infer important information on the complex histories of hybrids including selection, fitness in hybrids or the invaders, and admixture dynamics from range expansion.**”

Added text (Line 106-107): “Here, we evaluate the hypotheses that local expansion acts as resistance against intraspecies hybridization, and invasion success is heavily dependent on **both range expansion dynamics and the** invaders’ response to natural selection; both hypotheses have been often embedded in evolution studies.”

Added text (Line 113-116): We show that **if future hybridization events occur, they will occur at low frequencies; and that the introgressed pig genes should continue to dilute along the expansion of hybrids. Such results would provide evidence of demic diffusion in the wild, or that** natural selection favors the native genotypes, and that intraspecies competition or lack of survivability of the invasive species, has prevented the invaders’ naturalization.

Discussion

Amended paragraph (Lines 404-412);

“**The decrease of introgressed genes in the wild boars’ genomes with the increase in the distance from the invasion source (Table 1)** is likely to be caused from the native wild boar being most abundant in the evacuated zone [10] and the random mating tendencies of wild boar in this area, which in turn, increases the opportunity of backcross generations with wild boar and less chance of direct contact between remaining invasive pigs or other hybrids. **Particularly, our result may be an indication of demic diffusion [24] in the wild, in which the invader gene pool is progressively diluted along the expansion due to recurrent admixture with the native species.** This...”

Conclusion

Finally, in light of this alternative explanation the conclusion was rewritten with the selection idea less overclaimed.

“Here, using genetic markers, we demonstrated evidence of hybridization and introgression of invasive genes as a result of extrinsic environmental changes that reformed anthropogenic land use in Fukushima Prefecture in 2011. We found that there were likely successful introgression events in the evacuation zone following a sudden and large biological invasion, that thereafter spread and diluted across nearby wild boar at low frequencies. However, a massive introgression was not observed in this area and we propose two hypotheses for this result: 1.) The invasion success was heavily dependent on the invasive pigs attempt to naturalize in the wild area of Japan, and in this case, only a limited gene pool from invasive pigs could be passed on to the next generation of hybrids. 2.) An abundant native wild boar population and limited invasive pig population caused increasing introgression of wild boar genes into the invasive pigs and a decreasing introgression of invasive genes into the wild boar with the increase in the distance from the source of the invasion. In either case, if the invasion occurred in an environment that was not occupied already by high densities of wild boar, or if invaders had more advantageous survival traits for the wild, then maybe the invasive pigs would have had the same successful adaption as they had in other countries. We recommend that future studies assess the fitness of these hybrids and better characterize their ecological niche using

range expansion models and their ecological interactions. Such studies could determine if natural selection was invoked or if demic diffusion occurred after range expansion. Both scenarios suggest that the introgressed genes will eventually disappear in this area.”

Reviewer 1 Minor concerns:

-Lines 294-297; It would be better to show the coefficient of correlation (r) instead of the coefficient of determination (R²).

The coefficient of correlation (r) is now provided instead of coefficient of determination (R²), as suggested.

Lines 349-352; “A clear significant correlation ($r = 0.55$; $P < 0.001$) was observed among all wild boar subpopulations in the southern and northern regions. However, no significant correlation was observed ($r = 0.03$; $P = 0.131$) for wild boar within the Fukushima evacuated zones.”

-Line 329; “adaptation” instead of “adaption”

“adaption” was changed to “adaptation”

Lines 389; “...preferred local **adaptation** of the native species.”

-Lines 384-387; would you provide more details about why an increase in the population growth-rate of native wild boars would be a conservation concern?

More details of why an increase in population growth-rate of native boars would be a conservation concern were provided.

Lines 453-454; “and our evidence of introgression from pigs into the Japanese wild boars’ gene pool should be a concern for conservation biologists, **as increased abundancies of wild boar may affect plant cover and influence food web dynamics in their environment [46, 52].**”

46. VerCauteren KC, Beasley JC, Ditchkoff SS, Mayer JJ, Roloff GJ, Strickland BK, Schlichting PE, Beasley JC, VerCauteren KC. 2019 *The Naturalized Niche of Wild Pigs in North America*. (doi:10.1201/b22014-6)

52. Bueno CG, Alados CL, Gómez-García D, Barrio IC, García-González R. 2009 Understanding the main factors in the extent and distribution of wild boar rooting on alpine grasslands. *J. Zool.* **279**, 195–202. (doi:10.1111/j.1469-7998.2009.00607.x)

To Reviewer 2,

Thank you very much for your instructive and useful comments. We appreciate the time you invested in reviewing our manuscript so carefully. We have thoroughly revised our manuscript in response to your remarks. Especially for the analyses and presentation. We think your comments greatly improved the manuscript and we hope the revised version will now be reconsidered for your review.

Major Concerns:

-First of all, the writing style is far from appropriate, with a number of unclear sentences throughout the ms

We apologize for the inappropriate writing style and unclear sentences. We have carefully proof read the manuscript and attended to each of the highlights in your attached pdf file. We

hope that the revised manuscript, which followed the edits and comments of the Editor, Associate Editor, and Reviewers, has made the presentation and writing appropriate for publication.

-the use of abbreviations (SSRs) less common in this field compared to other ones (STRs)

While the use SSRs as abbreviations may be less common in certain fields, it does not necessarily mean the use of the abbreviation is inappropriate. In our case, the use of SSR marker is more common in the field of Ecology, while the use of STR marker is likely more common in Forensic DNA analysis. A quick google scholar search of either term has the similar number of results. We were happy to amend the manuscript and we now have used the abbreviation (STR) throughout the manuscript as suggested by the reviewer. In addition, we have amended the Supplementary file as well.

-the use of different concepts (introgression vs hybridization) not always used in the most appropriate way

We understand that introgression and hybridization are two different concepts and are not synonyms. Throughout the manuscript we have corrected and proof-read the use of each as defined by the review Twyford and Ennos, 2012 (Twyford AD, Ennos RA. 2012 Next-generation hybridization and introgression. *Heredity (Edinb)*. **108**, 179–189. (doi:10.1038/hdy.2011.68) We have also stated the following line in the materials and methods for clarity to readers:

Materials and methods

Added and amended text (Line 214-219); “For clarity, terms used in this manuscript are defined as follows. First, ‘hybridization’ refers to mixing between evolutionarily distinct lineages, i.e., mtDNA inheritance, whereas ‘introgression’ refers to gene flow between species as a consequence of hybridization, i.e., STR admixed genotypes. In this study, ‘hybrid wild boar’, or wild boar with hybrid ancestry, were determined based on the inherited pig lineage haplotype from mtDNA, or an admixed individual based on STRUCTURE (see above), or both.”

-too numerous to be indicated (you will find them highlighted in yellow in the attached pdf).

We have carefully edited/revised all of the highlights in attached pdf. We have also explained our corrections to each highlight with line numbers following these comments below, under the heading, “Highlighted concerns (highlighted by the referee in manuscript):”

-Some info presented in the Discussion should be better anticipated to the Introduction (e.g. number of escaped pigs, farms likely to have originated them, etc.).

The information previously presented in the discussion has been moved to the Introduction as suggested by the reviewer.

(Previously in **Discussion**): Line 343-345; “...wild boar was expected in this study because of the recent unintentional release of about 30000 pigs into the wild following the Japanese Government ordered evacuations in 2011.” and Line 392-393; “Our results suggest that abandoned livestock farms, inside the initial Japanese Government ordered evacuation zone (20 km radius from the FDNPP), were the source of the invasive pigs and the initial contact zone of introgression,...”

Was added to **Introduction**: Line 87-90;

Added text "...(2) reduced anthropogenic disturbances because of human evacuations [4], which included approximately 300 km² of urban and agricultural lands [5]; and (3) challenges from a recent invasion of about 30,000 pigs (*Sus scrofa domesticus*), released from abandoned farmlands [17] causing population..."

-The fact that samples had been used in previous studies should be clearly stated in Materials and Methods - Sampling section.

The samples used in previous studies are not clearly stated in the Materials and Methods – sampling section as suggested by the reviewer.

Added text Line 145-148; “Among the samples, the samples from Fukushima, Miyagi, and Yamagata Prefectures, and 10 pig samples were previously used in Anderson *et al.* [26]; and the 25 samples from Ibaraki Prefecture were used in Nagata *et al.* [27], which both studies analysed the mitochondrial DNA (mtDNA) control region.”

-Only 10 pigs (from a local market, but not necessarily representative of the gene pool of the escaped ones) have been analysed and this can strongly limit your ability to detect hybrids.

In regards to the limiting sample size of our analysis, please see our response to the editor above. In brief, while we initially shared the concerns of our limited sample size, and its ability to detect hybrids, our re-analyses using n=6,7,8, and 9 pig samples compared to n=10, as suggested by the Editor revealed that our sample size is not a limiting factor. After this analysis we are confident in our data analysis with 10 samples, as results were consistent after n=7 (see Response Table 1). This indicates that our markers are able to successfully detect the rare shared alleles between the released pigs and wild boar populations in Fukushima, which would be the indication of hybridization. If the pigs were under a wild and random-mating scenario then likely a greater number of individuals would be needed to detect hybrids. In this case, the pigs were in a controlled and highly selective setting, thus increasing the number of individuals will most likely not change allele frequencies and shared/introgressed alleles between pig and boar still remain limited. We found that a few individuals admixed in some independent runs, but not in others. Thus, we have made new thresholds and definitions for hybrid vs “suggested hybrid.” This is also described earlier in the letter.

However, we do understand these concerns stated by your comment and that this is still a possible limitation to the study. So, now we present this limitation in the manuscript that the number of hybrids detected is likely an underestimate due to our sample size of pigs. We address this in the manuscript for readers in the **Discussion** (Line 379-383).

Added text (**Discussion**, Line 379-383);

“Here, our results may be underestimating the full extent of introgression or hybrid occurrences because mtDNA data can only infer about female lineages (i.e., a cross between a female pig and a male wild boar) and our limiting sample size of pigs may not necessarily be representative of the complete gene pool of escaped pigs for STR. It may also be the case...”

-The correct name for the software "Structure Harvester" is "Structure Harvester".

"Structure Harvester" was corrected to "Structure **Harvester**" throughout the manuscript and supplemental files.

-A main weakness is that you define thresholds for identifying pure vs. hybrid boars and pigs before having identified the optimal number of clusters.

We apologize that this was not better clarified in the manuscript. However, we did not define thresholds to identifying pure vs hybrid boars and pigs prior to determining the number of clusters. All the Fukushima wild boar samples and pigs were analysed using STRUCTURE and the Earl et al., 2012 and Evanno et al., 2005 methods were used to determine the optimal number of clusters ($K=2$). These results were then validated using a PCA analysis which also positioned the samples in two clusters. This is now stated in the manuscript and presented in the Supplemental information.

The appropriate number of clusters for samples was $K=2$:

Materials and methods Line 197-205;

“STRUCTURE HARVESTER [36] was used to calculate the probability of the data for each K ($\text{LnP}(D)$) and ΔK [37]. The probability of the STRUCTURE analysis data remained at 0 with each K except a high peak at $K = 2$ (see Fig S2, electronic supplementary material), and the highest ΔK was detected when $K = 2$. Taken together, $K=2$ was the optimal number and we retained $K = 2$ to identify the proportions of admixture in wild boar and pig hybrids. For the selected $K = 2$, we assessed the average proportion of the membership coefficient (Q_1) to the inferred clusters.”

[36] Earl DA, vonHoldt BM. 2012 STRUCTURE HARVESTER: A website and program for visualizing STRUCTURE output and implementing the Evanno method. *Conserv. Genet. Resour.* **4**, 359–361. (doi:10.1007/s12686-011-9548-7)

[37] Evanno G, Regnaut S, Goudet J. 2005 Detecting the number of clusters of individuals using the software STRUCTURE: A simulation study. *Mol. Ecol.* **14**, 2611–2620. (doi:10.1111/j.1365-294X.2005.02553.x)

Results Line 277-280;

“The most probable number of clusters that captured greatest proportion of the data in the STRUCTURE analysis was $K = 2$ (Fig. 2) when the $\text{LnP}(D)$ and the ΔK were evaluated (see Fig S2, electronic supplementary material).”

Results Line 290-294;

“PCA analysis of the STR data showed similar genetic clustering and separated the data set into the two genetic clusters, with the identified hybrid wild boar positioned separately from the wild boar cluster, trailing off in the direction of the pig cluster (see Fig S3, electronic supplementary material).”

-Similarly, the analyses of geographical "populations" only based on the administrative borders lacks of significance and complicates the interpretation of data by the reader compared to the genetic clusters identified by Structure.

The use of geographical "populations" was used only as reference to originally identify each population based on sampling methods by prefectural hunters. We also did the analysis this way to determine genetic diversity among prefectures during the period prior to the FDNPP accident and evacuations. We agree with the reviewer that this additional analysis complicated the interpretation of the data for the readers. Thus, the analysis using geographical populations based on administrative borders was removed from the manuscript as its results did not impact the main conclusions or arguments we present. Now, the main genetic analyses used the appropriate geographical clusters as identified by STRUCTURE. We believe this will help interpret the data by the reader and appreciate the reviewer pointing this out.

Deleted text 242-245, "~~The 202 pure wild boar samples were originally classified into four populations according to their geographical locations: Yamagata Prefecture (Pop1), Miyagi Prefecture (Pop2), Fukushima Prefecture (Pop3), and Ibaraki Prefecture (Pop4), then grouped according to the assigned cluster determined by STRUCTURE for genetic analyses.~~"

Moreover, it is not clear why in some analyses you considered a sub-optimal number of clusters (3 vs. 2).

We apologize for the lack of clarity to determine the optimal number of clusters for our genetic analyses. Here, as stated above, we agree that this was not the best interpretation of the results. We have removed the analyses with populations by administrative borders and now present results based on STRUCTURE. We also removed analyses by using the sub-optimal number of clusters.

This is now clearly stated in the manuscript Line 195-198 and Supplemental info;

"To estimate the number of clusters (K), 10 independent runs with $K = 1-10$ were performed using 10^6 Markov Chain Monte Carlo (MCMC) iterations following a burn-in of 10^5 iterations. STRUCTURE HARVESTER [36] was used to calculate the probability of the data for each K (LnP(D)) and ΔK [37]."

Additionally, the appropriate number of clusters across all samples was $K=2$, but we showed $K=3$ as well. We did this here, because the STRUCTURE analysis of all individuals based on 24 loci, the probability of the data (LnP(D)) increased progressively with $K = 2$ and $K = 3$ and reached a plateau at $K = 3$ (below). As the highest ΔK was detected when $K = 2$ and the second highest value was detected for $K = 3$, so we described both K 's as is often done in other studies (i.e., Tsuda et al., 2017 Y. Tsuda, V. Semerikov, F. Sebastiani, G.G.M.L. Vendramin Multispecies genetic structure and hybridization in the *Betula* genus across Eurasia Mol. Ecol., 26 (2017), pp. 589-605). Additionally, the genetic analysis across geographical locations was removed and we know discuss results based on STRUCTURE.

There also may have been confusing to readers because when we did the hybrid analysis, we wanted to only include the samples collected from the period after the accident. This is also better stated in the manuscript.

Added text Line 182-183; "...samples from the period prior to the 2011 evacuations and outside Fukushima Prefecture were excluded."

To check that this was indeed the optimal number of clusters, PCA was used to confirm our STRUCTURE result. These are in supplemental information.

-The difference between the average pig content at different distances from the disaster zone was not tested for significance.

This was not tested for significance, but it was graphed and now provided as a new Figure in the manuscript (**Fig. 4**). However, given the limited number of hybrids and contingencies from sampling protocols based on governmental culling programs, we opted not to test for statistical significance as the result would merely reflect the sample size. In addition, majority of the inferred admixed hybrid individuals were within the initial evacuation zone <20 km from the FDNPP site. Because of this, the statistical analysis would not be appropriate here.

Added Figure 4: The estimated proportion of the membership coefficient (Q_1) plotted against the distance from the Fukushima Dai-ichi Nuclear Power Plant. Only wild boar with the maternally inherited pig haplotype or was an admixed individual were used. The mtDNA haplotype is indicated by color, green = maternally inherited pig haplotype and black = typical wild boar haplotype.

Moreover, given the availability of samples also across time, I would expect data from different years to be compared in order to better trace the expected dilution effect.

If you may, please also see the section addressed to the Associate Editor, where there was a shared interest/concern to account for the year sampling and age of the sampled individuals in our analyses. In brief, we would like to mention that the estimated age data coupled with genetic data, for a large mammal in the wild, is unique and we thought it would add to the manuscript. So, prior to initial submission we performed the statistical analyses, however we felt that while these analyzes may provide new perspectives, they are probably vulnerable to the limited number of hybrids detected by STRs and do not affect the main conclusions of this paper. In other words, we felt that 1., since sample locations and collected year is limited by the hunter's convenience and not of biological reasons (number of populations is not change) and 2., the age of boar would not make significant difference due to the random mating of populations (Fig S7) and quick maturity (less than one year) and birth periods for Japanese wild boar (see Kodera Y, Takeda T, Tomaru S, Sugita S. 2012 The estimation of birth periods in wild boar by detailed aging. *Mamm. Sci.* **52**, 185–191. (<https://doi.org/10.11238/mammalianscience.52.185>)). Thus, we felt the outcome of these analyses were not appropriate for presentation.

In hope to improve the manuscript in light of these shared concerns/interest we now:

- 1.) Subtracted the estimated age of the boar from the sampling month-year to approximate the period of birth, henceforth referred to as “birth year” of each hybrid boar. In other words, if a hybrid boar was sampled in July 2016 and was estimated to be 52 weeks old (~1 year) then the approximate birth year was July-2015. This new approximate birth year may help account for the lack of biological significance from year sampled. Using this birth year and pig content, we were able to provide some evidence of the dilution effect, as also mentioned by the second reviewer’s comment. However, there remains a poor correlation between Q_1 and birth year ($r = 0.05$), likely due to the small number of hybrids detected. The other possible reasoning for this is because, while our markers are able to detect hybrids, the released pigs may have been the progeny of prior hybrids due to the artificial selection/mixing of genes from boar to pigs by the farming industry as evidenced of Asian boar haplotypes in our pig samples (Anderson D, Negishi Y, Toma R, Nagata J, Tamate H. 2020 Robust microsatellite markers for hybrid analysis between domesticated pigs and wild boar. *Genet. Resour.* **1**, 29–41. (doi:10.46265/genresj.BNHB8715)). In this case, pig alleles are hidden in shared alleles or lost by backcross generations, and our markers cannot detect such hybrids.
- 2.) We now provide the age of hybrid boar and “birth year” in **Table S1**, and these data can be used for future reference.

We also added the text below:

Added text Results Line 309-312; “Using the approximated birth year of hybrid wild boars identified by STR analysis, five hybrids were approximated to be birthed in 2016 that ranged from 0.03 to 0.51 (average = 0.27); and five hybrids in 2018 that ranged from 0.04 to 0.33 (average = 0.21) (Table S1, electronic supplementary material).”

Overall, I think the manuscript could be highly improved in terms of data analyses, interpretation and style.

We truly appreciate your comments and edits. We have responded to each comment and reperformed data analyses, interpretation, and style of the manuscript. We do think these comments greatly improved the manuscript and hope that you will now find it appropriate for publication.

Below we have also responded to all of your highlights in the attached manuscript provided from the and how we have modified the paper.

Highlighted concerns (highlighted by the referee in manuscript):

Abstract

-Line 20; “Expansion changes”

Response Line 20-21; “Expansion changes” was changed to “**species** expansion”

-Line 27-28; “selection against dilution”

Response Line 28; “selection against dilution” was changed to “selection **against dilution** of the native gene pool”

-Line 30-31; “offspring from invasive lineages”

Response Line 31-32 “offspring from invasive lineages” was changed to “offspring from **hybrid** lineages”

-Line 32; “high intraspecies densities”

Response Line 33; “high intraspecies densities” was changed to “intraspecies **competition**”

Introduction

Line 44-46; “Of these extrinsic environmental changes and animal resilience efforts, long-lasting perturbations, including invasive species, expansion changes, and sudden fluctuations in population densities, could considerably impact ecology and evolution of many species.”

Response Line 45-49; Sentence was changed to the follow, “**Such extrinsic ecological changes and conditions could cause long-lasting perturbations and challenges, including invasive species, species expansion, and sudden fluctuations in population densities which may considerably impact evolution.**”

Line 54-55; “High densities of various species from an increased animal abundance in the evacuated area..”

Response Line 58-59; sentence was changed to “**Increased animal abundance of various species in the evacuated area might cause** new pressures...”

Line 58; “predatory populations saw their own population boom”

Response Line 62-63; sentence was deleted and changed to, “...even declined as predatory populations **and disease increased.**”

Line 64; “on the selection pressures of forced adaption.”

Response Line 69; “on the selection pressures of forced adaption” was changed to “...**the adaptive process.**”

Line 74; “invasive species”

Response Line 80’ “invasive species” was changed to “...**of influenced** species in these landscapes”

Line 77; “intraspecies population”

Response Line 83; “intraspecies” was deleted... “...suddenly introduced **intraspecies** population...”

Line 84; “Surely :

Response Line 93; “Surely” was changed to “**Presumably**”

Line 89; “wild boar, may”

Response: Comma was deleted.

Line 90-92; “So, what should be expected for native wild boar from such intraspecies hybridization and what may be inferred by this result for local adaptation versus an invasive species introgression success or naturalization?”

Response Line 97-101; sentence was rewritten;

“**Thus, investigating the influence of intraspecies hybridization and introgression here, a natural population influenced by a recent large invasion, could infer important information on the complex histories of hybrids including selection, fitness in hybrids or the invaders, and admixture dynamics from range expansion**”

Line 99; “predict”

Response Line 112; “predict” was changed to “**evaluated**”

Line 100; “locally adapted”

Response Line 116; “locally adapted” was replaced with “**the native genotypes**”

Materials and Methods

Line 116’ “evidenced of increased...”

Response Line 131; “evidenced of increased” was changed to “evidenced **by** increased...”

Line 120; Reviewer comment: “specify whether the samples were already used for previous studies”

Response Line 145-148; sampled previously used are now specified; “**Among the samples, the samples from Fukushima, Miyagi, and Yamagata Prefectures, and 10 pig samples were previously used in Anderson *et al.* [26]; and the 25 samples from Ibaraki Prefecture were used in Nagata *et al.* [27], which both studies analysed the mitochondrial DNA (mtDNA) control region.**”

Line 126; “Ten”

Response Line 142; “Ten” was not changed.

Line 142; “nSSR”

Response: “nSSR” was changed to “**STR**” throughout the manuscript and supplemental file.

Line 144; “selected markers”

Response Line 167; “selected markers” was deleted

Line 164; “for hybrid analyses, mtDNA and nSSR”

Response Line 189-191; “for hybrid analyses, mtDNA and nSSR” was deleted.

Line 172; “HARVESTOR”

Response “HARESTOR” was changed to “**HARVESTER**” throughout the manuscript and supplemental file.

Line 174; “assignment probability”

Response Line 202-205; “assignment probability” was deleted. This was also rewritten, “**For the selected $K = 2$, we assessed the average proportion of the membership coefficient (Q_1) to the inferred clusters.**”

Line 175-176; “In this case, a Q_1 value of one was considered to belong to pig and a Q_1 of less than 0.01 belonged to native wild boar”

Response Line 202-211; This sentence was rewritten.

“**For the selected $K = 2$, we assessed the average proportion of the membership coefficient (Q_1) to the inferred clusters. We assigned each individual to the inferred clusters, using a conservative threshold $Q_1 \geq 0.99$ for the assignment of individuals genomes to the pig cluster, or, $Q_1 < 0.01$ to the wild boar cluster. Admixed individuals were jointly assigned to the two clusters and were considered to have possible hybrid ancestry or were the offspring from ancestors that were a cross between wild boar and pig.**”

Line 185-186; “We therefore use the terms hybridization and introgression interchangeably, even though they are not considered synonymous.”

Response Line 219-221; This sentence was deleted.

Line 189; “view”

Response Line 221; “view” was changed to “evaluate”

Line 191; “indicated by haplotype”

Response Line 226; “indicated by haplotype” was changed to “**based on** haplotype”

Line 197; “0.01,”

Response Line 217-218; “0.01” was rewritten

Line 234-235 “...and **assigned to the inferred wild boar cluster** were used....”

Line 204; “four populations according to their geographical locations:”

Response: This has now been deleted.

Line 219; “main northern cluster as identified by STRUCTURE”

Response Line 258 “main northern cluster as identified by STRUCTURE” was changed to “**main**-northern cluster as identified by STRUCTURE”

Results

Line 234; “which clearly”

Response Line 274; “which clearly” was rewritten to “...mtDNA ~~which clearly and~~ distinguished...”

Line 238-239; “the range of the two species was divided into two clusters with 15 wild boar admixing.”

Response Line 280-282; The sentence was rewritten, “At $K = 2$, all individuals were assigned to two clusters with 10 admixed individuals in 100% of the independent runs and an additional five admixed individuals in more than 50% of the runs.”

Line 254-256; “The proportions of hybrids compared to all the sampled wild boar in both the 20 km and 20-40 km radius zones were high at 16% and 25%, respectively.”

Response Line 301-302; This sentence was deleted.

Line 259-260; “i.e., 0.1 and 0.02”

Response Line 302-304; The sentence was rewritten, “~~Additionally, the shared ancestry to pig was higher for hybrids in the initial 20 km radius evacuation zone or, in other words, there was evidence of higher introgression inside the evacuated zones (Fig. 4).~~”

Line 289; “overlapped position coordinates”

Response Line 343; “overlapped position coordinates” was changed to “...~~tended to overlap~~ position coordinates...”

Discussion

Line 309-311; “The two hybrids sharing both the pig maternally inherited haplotype, and a STRUCTURE result indicating a proportion of the hybrid genome originated from a pig population, further support a scenario of introgression from invasive pigs.”

Response Line 364-367; sentence was rewritten, “~~The observed maternally inherited pig haplotype in two of the admixed individuals, further support a scenario of hybridization.~~”

Highlight and comment: Should be moved to discussed in introduction. Line 317-319; “A higher frequency of hybrid wild boar was expected in this study because of the recent release of about 30000 pigs into the wild following the Japanese Government ordered evacuations in 2011.”

Response Line- As described earlier in this letter, the estimated number of released pigs is now discussed in the **Introduction** as recommended by the reviewer.

Line 331; “positive selection”

Response Line 391; “positive” was deleted.

Line 333; “and it is unlikely”

Response Line 395; “and it is unlikely” was deleted and sentence rewritten, “~~We suggest that the pig mtDNA or introgressed pig genes do not have higher fitness than native wild boar in the wild based on decreasing hybrid frequency with the increase in the distance.~~”

Line 342; “The reduction of Q1 values”

Response Line 404, This sentence was rewritten, “**The decrease of introgressed genes in the wild boars’ genomes with the increase in the distance from the invasion source** (Table 1)...”

Review comment “across years/ages?”

Response- Please see the response earlier in the letter regarding the comment.

Line 358; Reviewer comment “where were them?” in regards to location of farms

Response: Unfortunately, we do not have the exact location of the pig farms as we understand that this information would be very useful, but only know that approximately 300 km² of urban and agricultural lands were within the evacuation zone. We can assume these were the source of the pigs and our data can infer that the evacuation area was the source of the invasion of pigs.

We now state more information about the evacuation and farms in the **Introduction** Line 86-88; “...reduced anthropogenic disturbances because of human evacuations [4], **which included approximately 300 km² of urban and agricultural lands [5];and...**”

Line 362; “a pig”

Response Line 428 “a pig” was amended to “**pigs**”

Line 368; “dispersing greater distances”

Response Line 432 Sentence was deleted

Line 379; “climates continue”

Response Line 446; “climates continue to warm” was changed to “climates warm”

Line 388; “inbreeding”

Response Line 455; “inbreeding” was deleted.

Conclusion

Line 415-416; “that local adaption acted”

Response 462-480; The conclusion was rewritten.

Line 420; “bias to the local adapted”

Response 462-480; The conclusion was rewritten.

“Here, using genetic markers, we demonstrated evidence of hybridization and introgression of invasive genes as a result of extrinsic environmental changes that reformed anthropogenic land use in Fukushima Prefecture in 2011. We found that there were likely successful introgression events in the evacuation zone following a sudden and large biological invasion, that thereafter spread and diluted across nearby wild boar at low frequencies. However, a massive introgression was not observed in this area and we propose two hypotheses for this result: 1.) The invasion success was heavily dependent on the invasive pigs attempt to naturalize in the wild area of Japan, and in this case, only a limited gene pool from invasive pigs could be passed on to the next generation of hybrids. 2.) An abundant native wild boar population and limited invasive pig population caused increasing introgression of wild boar genes into the invasive pigs and a decreasing introgression of invasive genes into the wild boar with the increase in the distance from the source of the invasion. In either

case, if the invasion occurred in an environment that was not occupied already by high densities of wild boar, or if invaders had more advantageous survival traits for the wild, then maybe the invasive pigs would have had the same successful adaptation as they had in other countries. We recommend that future studies assess the fitness of these hybrids and better characterize their ecological niche using range expansion models and their ecological interactions. Such studies could determine if natural selection was invoked or if demic diffusion occurred after range expansion. Both scenarios suggest that the introgressed genes will eventually disappear in this area.”

Appendix B

Response to referees' letter

Journal Name: Proceedings of the Royal Society B

Journal Code: RSPB

MS Reference Number: **RSPB-2021-0874**

Title: Introgression dynamics from invasive pigs into wild boar following the March 2011 natural and anthropogenic disasters at Fukushima

Dear Professor Loeske Kruuk,

First, we would like to again thank the Associate Editor, the referees, and you for the new decision letter sent on 28th May 2021 regarding assessment of our manuscript (**RSPB-2021-0874**) entitled "Introgression dynamics from invasive pigs into wild boar following the March 2011 natural and anthropogenic disasters at Fukushima" by Anderson, Donovan; Negishi, Yuki; Ishiniwa, Hiroko; Okuda, Kei; Hinton, Thomas G.; Toma, Rio; Nagata, Junco; Tamate, Hidetosh B.; Kaneko, Shingo. We have thoroughly revised our manuscript in response to the new suggestions and comments from the Referees. The details of improvements, revisions, and how we responded to their individual comments are presented in the remainder of this letter. In addition, we hope we have adequately addressed the two bigger concerns about the few hybrids detected with both markers, and transparency of our selected STRs. We hope that you will find the revised version suitable for publication in the Proceedings of the Royal Society B.

Please note that Line numbers will refer to the attached file 'manuscript marked with changes.'

To the Associate Editor Board Member,

Again, thank you very much for your coordinating roles as an associate editor. We appreciate the time you and the referees invested in reviewing our manuscript. We have responded to all of the comments by Referee 2 and Referee 3, and have revised our manuscript accordingly. All of the comments were useful and improved the manuscript.

Comments to Author:

The reviewers both feel that this paper is improving and is highly likely to make an important contribution to this field. There are a number of small points and two large conceptual ones remaining. The two big ones are: 1) address why so few of their hybrids were detected by both markers and 2) be more transparent about the power of their STRs, by looking at their population specific allele frequencies and reporting CIs. The reviewers also identified a number of more minor issues.

We are pleased to hear that the paper is improving and is thought to make an important contribution to the field. All of the small points mentioned have been addressed and the manuscript was revised accordingly. We have also addressed the two large conceptual comments, regarding 1) the small number of hybrids detected by both markers and 2) the transparency/power of our STRs. If you may, please refer to the portion of the letter where we address these comments to Referee 3 in more depth regarding the two larger conceptual issues.

In brief, we believe that there may have been confusion that the power or allele frequencies of the STRs were not tested or reviewed. However, we selected these robust markers based on

Anderson et al (2020) where we reviewed 52 STR markers in a primer note manuscript (see **additional note** below our revised passage).

In light of the comments, we added the allele frequencies for populations and power of SSR markers Tables (**Table B and Table C**) to the letter below addressed to Referee 3. In the revised manuscript we added **Table C** (in the electronic supplementary file Table S2) and we added CIs to Table S1 as suggested. However, we did not add **Table B** to the manuscript because we felt like the prior primer note manuscript explains the results of the selection of markers in more detail (again see **additional note** below our revised passage). We added the following sentence for clarity to readers on where to find the information:

L157-159 “A total of 24 STR loci were selected and genotype for our study populations **based on the allele frequencies and the amplification for each of these markers in pure species individuals** (see Anderson et al [28]).

We have also added text to the manuscript regarding the power, and transparency of the markers and why so few hybrids were detected by both markers as suggested by the Referee 3. The revised paragraph below was added after we verified the power of our markers (**Table C** and added to electronic supplementary file Table S2).

Added sentence Line 334-337: “It may also be the case that our low pig ancestry results suggest a scenario that hybrids in this study were probably the offspring of one or more backcross generation of pure-wild boar, which would halve the number of introgressed pig alleles at every generation. **While the selected STR markers consistently detected third backcross (BC2) generation hybrids (about 88 %, see electronic supplementary material, Table S2), it is feasible to assume a fourth or more backcross generation hybrid would be beyond the marker’s detection ability.**”

Revised paragraph Line 370-384: “**Despite such a sudden and large invasive force from pigs, our data likely suggest that most pigs failed to naturalize in the wild and survival of male pigs may have been higher. Here, we observed minimal selection of the pigs’ invasion from the remaining invasive pig mtDNA haplotype in wild boar and the reduction of pig alleles, which is more pronounced in the STR loci because they are a mixture from both sexes. However, our results also detected only two individuals as hybrids by both the mtDNA and the STR markers. Previous studies have also pointed out inconsistencies in hybrid determination by mitochondrial and nuclear markers [15]. Given that our selected markers consistently detected third generation backcrosses (see electronic supplementary material, Table S2), we can speculate that female pigs and maternal offspring had difficulty surviving. If pig farms tended to raise more female pigs than male pigs, then this survival bias may explain the lack of concurrence in hybrid determination by mitochondrial and nuclear markers, and the seemingly unnatural distribution of Q_1 values in mitochondrial hybrids. Additionally, in the U.S., male wild pigs tended to have higher survivability contributed to larger mass at birth [45]. Thus, it may be possible that male wild pigs or their paternal lineages in this area may have been more successful than female pigs.**”

Additional Note: We selected the markers specifically for this larger based study for hybrid analysis between pigs and boar by removing markers that did not successfully amplify with the sub-sample set, or showed common alleles between pigs and boar from the period prior to the 2011 events. The selected markers’ allele frequencies between each individual population are fully explained in that manuscript. We felt that publishing the selection of these markers and the allele frequency results were best as a separate manuscript, as the use of these markers was

directed for users of STRs as a robust cost-efficient analysis. The Anderson et al. 2020 manuscript evaluated these markers by assessing common or shared alleles between the Fukushima wild boar during the period after 2011, wild boar from the period prior to 2011, and domestic pigs. The robust markers successfully found shared alleles between the Fukushima wild boar and pigs, which suggests that the alleles were introgressive through mixing of pigs and wild boar during the period after 2011 evacuations and Fukushima disasters. Thus, we went forward with the selected markers for this study to evaluate the hybridization event occurring here.

--Anderson D, Negishi Y, Toma R, Nagata J, Tamate H. 2020 Robust microsatellite markers for hybrid analysis between domesticated pigs and wild boar. *Genet. Resour.* 1, 29–41. (doi:10.46265/genresj.BNHB8715)

To Referee 2,

Comments to the Author(s).

The authors tried hard to improve their ms, partly reaching this goal. Still, there are several passages to be further improved.

My coauthors and I would like to thank this Referee for their additional critique of our manuscript. We again found their comments useful and thank them and time invested into assessing our manuscript carefully. We do apologize that the previous revision did not fully improve our manuscript. Below, we have addressed all of the comments and revised the manuscript accordingly. We feel like the new comments helped the manuscript and we hope that the amended passages are now suitable for the Referee and publication.

L. 29-30: highly speculative here

The speculative nature of the abstract was rewritten to be more informative. In addition, Line 29-30, the word “we predict” was changed to “we speculate...”. While we understand more analysis is needed to verify our speculation, we believe the current data, as presented, show a decreasing tendency of pig ancestry in the wild boar population in this area due to on-going backcross. Therefore, we feel it is best to maintain our sentence of this speculation.

Original: “Concurrently, we show how invasive pigs failed to naturalize in the same landscapes and, despite evidence of successful hybridization between pigs and native wild boar in this area, in future offspring, selection of the native gene pool or range expansion dynamics inhibit long-term introgression. Using our data, we predict that introgressed alleles will continue to decrease at each generation and only maternally inherited organelles will remain.”

Revised Line 28-31: “Concurrently, we show evidence of successful hybridization between pigs and native wild boar in this area, **however** in future offspring, **the pig legacy has been diluted through time.** **We speculate that** the range expansion dynamics **inhibit** long-term introgression and introgressed alleles will continue to decrease at each generation and only maternally inherited organelles will persist.

L. 37: "overall loss"

“loss” was changed to “overall loss” as suggested.

L. 52: repetition

Repetitive part of the sentence was deleted as suggested.

Line 56 “~~Anthropogenic pressures were suddenly removed from landscapes within the~~ In the evacuated areas, wildlife census”

L. 79: i don't think that in the evacuation area you can consider multiple "populations"

We agree with the referee about this consideration and have deleted both mentions of the word “populations”

L. 80-81: please report any available estimate, in order to be comparable with the number of escaped pigs

As suggested, we now provide an estimate from a Fukushima Prefectural report:

Line 84-86 “((1) sudden population expansion as evidenced by greater population abundance [10] ~~and population growth in the area (estimated 49,000 to 62,000 boar from 2014 to 2018 [17]);~~”

L. 96-97: unclear, please rephrase

The sentence was rephrased as suggested.

Old: “Here, we evaluate the hypotheses that local expansion acts as resistance against intraspecies hybridization, and invasion success is heavily dependent on both range expansion dynamics and the invaders’ response to natural selection; both hypotheses have been often embedded in evolution studies”

Revised 101-104: “Here, we evaluate the hypotheses that range expansion acts as resistance against hybridization, and that invasion success is dependent on the invaders’ response to naturalization; both hypotheses have been often embedded in evolution”

L. 138: delete "studies"

“studies” was deleted.

L. 178: please motivate why you used correlated allele frequencies even dealing with a wild population and a domestic one

We decided to use the model of correlated allele frequencies for STRUCTURE because this implemented model says that frequencies in different populations are likely to be similar (probably due to shared ancestry). This model often improves clustering for closely related

populations, in this case pig and boar, which have been known to share alleles (Falush D, Stephens M, Pritchard JK (2003) Inference of population structure using multilocus genotype data: linked loci and correlated allele frequencies. *Genetics*, 164, 1567–1587). Additionally, in Asia domesticated pigs are often bred with wild boar for certain genetic traits. Thus, we felt this model was better suited than the independent or noncorrelated allele frequencies models to assess hybridization as pig and boar are mixing and also because they have common alleles in our study area. Other studies have used this model to assess hybridization/introgression among wild and domestic species as well [1–5].

1. Randi E. 2008 Detecting hybridization between wild species and their domesticated relatives. *Mol. Ecol.* **17**, 285–293. (doi:10.1111/j.1365-294X.2007.03417.x)
2. Godinho R *et al.* 2011 Genetic evidence for multiple events of hybridization between wolves and domestic dogs in the Iberian Peninsula. *Mol. Ecol.* **20**, 5154–5166. (doi:10.1111/j.1365-294X.2011.05345.x)
3. Tsuda Y, Semerikov V, Sebastiani F, Vendramin GG, Lascoux M. 2017 Multispecies genetic structure and hybridization in the *Betula* genus across Eurasia. *Mol. Ecol.* **26**, 589–605. (doi:10.1111/mec.13885)
4. Kidd AG, Bowman J, Lesbarrères D, Schulte-Hostedde AI. 2009 Hybridization between escaped domestic and wild American mink (*Neovison vison*). *Mol. Ecol.* **18**, 1175–1186. (doi:10.1111/j.1365-294X.2009.04100.x)
5. Hansen Michael M. and Mensberg Karen-Lise D. 2009 Admixture analysis of stocked brown trout populations using mapped microsatellite DNA markers: indigenous trout persist in introgressed populations *Biol. Lett.* 5656–659 <http://doi.org/10.1098/rsbl.2009.0214>

Additionally, typically the risk of the correlated allele frequency model is overestimating the K, as shown in the manual, and our data/results were not at risk of this as we also analyzed only the wild population (Fig. 5), which would remove the pig cluster or the population that was quite divergent from the others.

L. 197: mtDNA inheritance can be representative of either hybridization AND introgression; the same applies to STR admixed genotypes. I'd keep both the correct definitions provided, but without referencing to these misleading examples

We agree with the referee with this point and apologize for the poor references that were included. As suggested by the referee, both correct definitions were retained and the misleading examples were deleted.

L. 199-201: again, a boar with pig mtDNA but 100% STR assignment to wild boar pop is an example of introgression

The sentence was changed as suggested.

Line 201-203 “In this study, **introgressed** or hybrid wild boar were determined based on the inherited pig lineage haplotype from mtDNA, or an admixed individual based on STRUCTURE (see above), or both.”

L. 219-221: please consider whether it is worth using the samples collected before 2011 separately in order to evaluate possible genetic drift

Yes, as suggested, we did consider whether it was worth using the samples collected before 2011 separately in order to evaluate possible genetic drift. However, our analysis of pure wild boar (Fig. 5), allele frequencies (see Anderson et al 2020) and previous table of genetic diversity (Table A was deleted following revisions of first letter after decision it cluttered the main points of the manuscript - but we provided below) indicate no genetic drift due to the similar allelic variation among before and after 2011 and similar H_O and H_E values in the Pop1, Pop2(Northern population before 2011), and Pop3 (Fukushima after 2011). Like referee 2, we expected genetic drift to be found in the wild boar northern populations collected prior to 2011 and the northern group collected after 2011. Surprisingly, all data show no indication of a genetic drift. The F -values were small and similar, indicating no genetic drift between northern populations (Fig 5). This suggests that the northern group collected prior to 2011 were likely already separated from the southern group collected before 2011. This result can be explained likely from the rapid increase in individuals in this area, before 2011 expansion of distribution area had reported. Additionally, we feel like this information is already provided in the manuscript given its current presentation. The limited sample size of wild boar collected prior to the accident ($n = 42$), we feel, does not warrant additional presentation in the manuscript.

Table A (Note: deleted following revisions of 1st letter on March 25th, 2021, but provided here for clarity to our response). Polymorphism measurements and genetic diversity of four pure-wild boar populations in four prefectures in Japan based on the analysis of 24 microsatellite loci. Pop1, Pop2, Pop3, and Pop4 refer to wild boar populations from Yamagata, Miyagi, Fukushima, and Ibaraki Prefectures, respectively. Note that the northern group at $K=2$ was consisted of Pop1, Pop2 and Pop3, and the southern group Pop4. n = number of sampled individuals; N_A , mean number of alleles; R_s , allelic richness; H_O , mean observed heterozygosity; H_E , mean expected heterozygosity; F_{IS} , inbreeding coefficient.

Population	n	N_A	R_s	H_O	H_E	F_{IS}
Pop1	7	3	3	0.46	0.46	0.056
Pop2	10	3	2.7	0.38	0.42	0.15
Pop3	160	6	2.8	0.41	0.42	0.029
Pop4	25	4	3.1	0.43	0.49	0.15

L. 247-248: please postpone after you have defined the two reference populations in the following sentence

The sentence was postponed until after the two reference populations were defined (Line 250-253)

L. 268-270: what about the other 21 hybrids? Why didn't you include them in this comparison? You have these data available, please use it

Similar to Referee 2, we also thought that we should use the 21 hybrids in the analysis for comparison. However, we ultimately found no purpose for this STRUCTURE analysis result.

The purpose of our pure-boar STRUCTURE analysis was to determine admixture and gene flow of pure-wild boar to determine possible dispersal or movement patterns of hybrids in the future. If we included the hybrids, we are confident that STRUCTURE could recognize the hybrid cluster, so the resolution of the pure-boar population analysis would decrease. Additionally, we now provide the PCA analysis results using all samples in the supporting electronic supplementary file. Here, we can also see the clustering tendencies of all samples, and the evaluation of clustering of samples from the period prior to and after the 2011 events.

We do now provide genetic clustering using PCA analysis with all samples (supporting electronic supplementary Fig S5) – Shown below:

L. 273: typo

Typo was corrected.

“Birthed in” was changed to “born”

L. 278-279: Please explain better (as you did in the answer to reviewers)

The explanation was revised to match that of the first response letter to referees as suggested.

Line 279-283 “The $\ln P(D)$ increased progressively from $K = 2$ to 3 and then plateaued after $K = 3$ (electronic supplementary material, Fig S4). Additionally, the highest ΔK was detected when $K = 2$ and the second highest when $K = 3$. Thus, the most likely number of clusters was $K = 2$ followed by $K = 3$. At $K = 2, \dots$ ”

L. 286-289: Please, highlight where pre-2011 individuals fall among these clusters

We now provide the PCA analysis with all samples and highlight the pre-2011 individuals as suggested. This is now provided in the electronic supplementary material. In addition, we highlighted in the text.

Line 294-297 “In addition, the northern group wild boar collected from the period prior to 2011, suggested little genetic drift from the samples collected after 2011 ($F = 0.197$, Fig. 5), while the

the southern group samples diverged (Fig. 5, and see electronic supplementary material, Fig S5).”

L. 319: "pig ancestry" instead of "admixture"

As suggested, “admixture” was changed to “pig ancestry”

L. 328-329: as you stated in the response letter, bred pigs probably show limited diversity compared to wild boars, please be consistent

We apologize for the possible confusion of the word “limited” used in the letter and the manuscript. This has been revised throughout the manuscript. If we are correct, the Referee is referring to the sentence in the previous letter (March 25th) when we said, “it is likely that adding more pigs will not largely influence or change allele frequencies and the shared alleles between pig and boar still remain limited in this area. This is likely the case because the pig populations are not under random mating, but a highly selective or controlled population in the agriculture setting.” While this is true, we did not intend to imply that the bred pigs showed limited diversity compared to wild boars. Other studies show that pig genetic diversity is quite high or similar to that of wild boar. In our study, 6 different haplotypes were detected in 10 samples in comparison to the 3 haplotypes (including the hybrid haplotype) detected for 191 wild boar. We have revised the sentence for clarity.

L. 331 (LINE 323-8 lines): "favoured" instead of "preferred"

This section was deleted/rewritten. Please see the next comment for revised section.

L. 333 (LINE 325-8 lines): "under" instead of "during"

This section was deleted/rewritten. Please see the next comment for revised section.

L. 333-339: all this part is unclear and/or the assumptions are not properly supported. Please, carefully review it and make it coherent with the following section (is the observed pattern of decrease of pig ancestry compatible with selection or with neutral dilution under random mating? Based on the current analyses, you have no chance to document selection unless you formally test for it

Here, we kindly disagree with the Referee. We do think the manuscript overstated the conclusion of “selection” without formally testing for it, so in response, we have made revisions accordingly and deleted the mention of “selection” or “natural selection”. We have made sure to not mention this in the abstract, results, or conclusion. However, we keep our revised passage about our speculation of survivability in the discussion following comments from other referees.

We also moved the order of the passages to provide readers with the observed pattern of decrease of pig ancestry compatible with neutral dilution under random mating first, as this is likely the

case. However, it cannot be ruled out that the selection or survivability of pigs in the wild had no role in the pigs' invasion. As referee 2 states later, 30,000 pigs is a large number. This means that most of the pigs likely died in the wild following their release. This is supported by a large number of haplotypes detected in pigs, but only 1 found in the wild. Additionally, following referee 3's comments, we found that our markers are relatively strong (high power, see **Table C** in letter response to Referee 3) in detecting hybridization in first generation hybrids, and second (BC1) and third backcross (BC2) generation hybrids. Our results show two individuals that were detected as hybrids by both the mtDNA and the STR markers. Originally mtDNA hybrid individuals were born, and distribution of Q should be similar to nSSR hybrids. But, these individuals cannot survive. Thus, however speculative, we have rewritten the unclear parts from L333-339 removing selection discussion, but discussing the speculation above.

Revised 371-385: “Despite such a sudden and large invasive force from pigs, our data likely suggest that most pigs failed to naturalize in the wild and survival of male pigs may have been higher. Here, we observed minimal selection of the pigs' invasion from the remaining invasive pig mtDNA haplotype in wild boar and the reduction of pig alleles, which is more pronounced in the STR loci because they are a mixture from both sexes. However, our results also detected only two individuals as hybrids by both the mtDNA and the STR markers. Previous studies have also pointed out inconsistencies in hybrid determination by mitochondrial and nuclear markers [15]. Given that our selected markers consistently detected third generation backcrosses (see electronic supplementary material, Table S2), we can speculate that female pigs and maternal offspring had difficulty surviving. If pig farms tended to raise more female pigs than male pigs, then this survival bias may explain the lack of concurrence in hybrid determination by mitochondrial and nuclear markers, and the seemingly unnatural distribution of Q_1 values in mitochondrial hybrids. Additionally, in the U.S., male wild pigs tended to have higher survivability contributed to larger mass at birth [45]. Thus, it may be possible that male wild pigs or their paternal lineages in this area may have been more successful than female pigs.”

Again, we do understand that this is speculative. But, based on comments from other referees we think it is an interesting point to keep in the discussion. So, we have revised accordingly. We also keep in mind that it is difficult to fully discuss/say due to the small number of samples and state this.

L. 366: "pig ancestry" instead of "admixture to the pig population"

“admixture to the pig population” was changed to “pig ancestry”

L. 377: "genetic mixture"  do you mean "gene flow"?

“genetic mixture” was changed to “gene flow”

L. 395: "genetically similar populations"  not clear

“genetically similar populations” was deleted.

“The greater abundance and higher densities of wild boar in the Fukushima evacuated area [10] coupled with our findings of ~~genetically similar populations and~~ hybridization from...”

L. 417: "hybridization" instead of "introgression"

“Introgression” was changed to “hybridization”

L. 420-423: (Really 411-414) all unclear

Please see our earlier comment about the unclarity or our speculations. In this case, the referee is referring to our speculative statement that the “invasion success was heavily dependent on the invasive pigs attempt to naturalize in the wild area of Japan, and in our case, only a limited gene pool from invasive pigs could be passed on to the next generation of hybrids”.

We have kept the speculative nature of the sentence; however, we have revised it to match the discussion and avoid the word of “selection”. Additionally, we mention the demic dilution hypothesis first and then we mention the other possible hypothesis of the poor survivability of pigs in the wild. Please see change below:

~~Deleted: and in this case, only a limited gene pool from invasive pigs could be passed on to the next generation of hybrids. 2.) An abundant native wild boar population and limited invasive pig population caused increasing introgression of wild boar genes into the invasive pigs and a decreasing introgression of invasive genes into the wild boar with the increase in the distance from the source of the invasion.~~

Revised Line 4450449 “However, a massive introgression was not observed in this area and we speculate two hypotheses for these patterns: 1) An abundant wild boar population caused increasing introgression of wild boar genes into the invasive pigs and a decreasing introgression of invasive genes into the wild boar with the increase in the distance from the invasion source. 2) The pig legacy passed on to the next generation hybrids was dependent on the ability for pigs to naturalize in the wild area of Japan.”

L. 423: i guess 30.000 is hardly definable as “limited”

We apologize for the use of the word “limited.” We agree with the reviewer that 30,000 pigs is large, and may not be limited. However, we did not intend the word limited in this way. Our intent was to say that the estimated number of pigs is finite. Thus, the limiting factor is the gene pool from pigs able to survive in the wild and capability to pass on to next generations. As we wrote, “a limited gene pool from invasive pigs could be passed on to the next generation of hybrids.” To make this clearer we have added a sentence saying most pigs likely could not survive in our revised passage.

Additionally, we deleted the word “limited” from L423 as suggested.

To Referee 3,

Comments to the Author(s).

The paper ‘Introgression dynamics from invasive pigs into wild boar following the March 2011 natural and anthropogenic disasters at Fukushima’ asks about the hybridization between wild boar and domestic pigs that might have occurred since the Fukushima melt down in 2011. I like this paper. It’s clear and well written and reasoned, and asks a really interesting question. I have a few points of clarification on some of the analyses, which, I hope are straightforward to address.

Thank you for your helpful comments. The few points of clarification on some of the analyses have been addressed below and in light of these comments, we were able to revise the manuscript to add additional discussion, specifically regarding possible reasons why few hybrids share both markers after verifying the power of the markers used. These revisions are below.

I would really like to see this paper address the power of the 24 STR markers that are used in the nuclear DNA analysis. Are these markers highly diverged, diagnostic or ancestry informative between wild boar and pigs? Or are they just polygenic and amplifying in both? I would describe the allele frequencies for each of these markers in pure species individuals.

We apologize that the paper did not better discuss the power/transparency of the markers further. We selected these robust markers based on Anderson et al (2020), a primer note manuscript, where we reviewed 52 STR markers in a sub-sample set. Across all markers, there were several that were polygenic and amplified in both wild boar and pigs. However, for this study, polygenic markers were not selected. We selected the markers specifically for our study population for hybrid analysis between pigs and boar by removing markers that did not successfully amplify with the sub-sample set, or showed common alleles between pigs and boar from the period prior to the 2011 events. The robust markers successfully found shared alleles between the Fukushima wild boar and pigs, which suggests that the alleles were introgressive through mixing of pigs and wild boar during the period after 2011 evacuations and Fukushima disasters. Thus, we went forward with the selected markers for this study to evaluate the hybridization event occurring here.

We have provided **Table B** below to show the allele frequencies for pure-wild boar from the northern group collected prior to 2011, which showed similar clustering/mixing to the Fukushima population, and pigs (see also **Table A** addressed to Referee 2 above).

--Anderson D, Negishi Y, Toma R, Nagata J, Tamate H. 2020 Robust microsatellite markers for hybrid analysis between domesticated pigs and wild boar. *Genet. Resour.* **1**, 29–41. (doi:10.46265/genresj.BNHB8715)

Table B: Allele frequencies for all populations by locus (%). The ‘northern boar pop’ consists of the northern boar populations (by PCA and structure analysis) from the period prior to the 2011 events, which these wild boars were closely related to the northern boar collected in the period after the 2011 events.

Here we can see that our markers can detect pig specific alleles, boar specific alleles, as well as shared common alleles.

Allele frequencies for all populations by locus (%)				
Loci	Northern boar pop		Pig	
	Boar specific	shared	Pig specific	shared
Sw632	79.4	20.6	40.0	60.0
S0090	0.0	100.0	45.0	55.0
Sw24	11.8	88.2	10.0	90.0
Swr1941	79.4	20.6	70.0	30.0
Sw857	8.8	94.1	55.0	45.0
S0228	55.9	44.1	80.0	20.0
Sw2008	100.0	0.0	100.0	0.0
Sw240	79.4	20.6	55.0	45.0
S0097	94.1	5.9	95.0	5.0
UMNp239	91.2	8.8	85.0	15.0
UMNp500	100.0	0.0	100.0	0.0
UMNp511	70.6	29.4	40.0	60.0
UMNp610	14.7	85.3	80.0	20.0
UMNp442	0.0	100.0	90.0	10.0
UMNp548	100.0	0.0	100.0	0.0
UMNp362	100.0	0.0	100.0	0.0
UMNp485	91.2	8.8	95.0	100.0
UMNp539	5.9	94.1	45.0	55.0
UMNp509	970.6	2.9	95.0	5.0
UMNp381	73.5	26.5	5.0	95.0
UMNp640	0.0	100.0	50.0	50.0
UMNp489	52.9	47.1	75.0	25.0
UMNp351	58.8	41.2	50.0	50.0
UMNp296	52.9	47.1	95.0	5.0

We did not add **Table B** of allele frequencies to the manuscript as we did not think it would improve the manuscript as it is described carefully in the earlier publication. However, we have better clarified where to find the information and transparency of the markers:

L158-160 “A total of 24 STR loci were selected and genotype for our study populations based on the allele frequencies and the amplification for each of these markers in pure species individuals (see Anderson et al [28]).

What really caught my eye is that 1) Table 1 appears to only have two individuals that were detected as hybrids by both the mtDNA and the STR markers and 2) the few ‘suggested hybrids’ (i.e., those indicated as hybrids in some runs of STRUCTURE, but not all) have substantially lower Q scores than the hybrids consistently detected by STRUCTURE. I can imagine a couple of explanations for these patterns.

The first is that the 24 markers are not powerful enough to detect later generation backcrosses consistently. Based on a quick search, I think that these animals have a generation time of 2- 3 years, so I estimate that around 3rd generation backcrosses could be common in this population. 3rd generation backcrosses have an average Q score of ~0.06. Do these 24 markers consistently detect those individuals? Could there have been even more generations of backcrossing elapsed since the Fukushima disasters?

I really have no idea, but this would be worth addressing. I think that the best way to do this would be to include credible intervals on the STRUCTURE plot (Figure 2), and in Table S1 beside the Q scores for all hybrids. Boecklen and Howard (1997) have a nice deterministic model describing how many markers are needed to detect backcrossing and Vähä and Primmer (2006) also describe this problem and use simulations to make suggestions. The manuscript hints a bit at this general problem on lines 319-321, but I do think it is addressable to some extent.

Based on this comment we evaluated the power of our STR markers (**Table C**) and added the CIs to Table S1 and **Table C** as Table S2 in the electronic supplement file. We found that our markers were consistent in detected the 3rd generation backcross (88%). Thus, we do believe it is possible that even more generations of backcrossing elapsed since the Fukushima disasters where we think is likely beyond detection ability of the 24 markers. This would also explain our suggested hybrids with inconsistent detection ability at average Q score of ~0.01. Additionally, we cannot be certain of the initial shared alleles with that of the released pigs involved in the early hybridization and the wild boar. Thus, these initial common alleles may reduce the resolution of indication of the true number of backcross generations of hybrids.

Please note the CIs for the hybrids are small due to the consistent detectability. Thus, we did not include these in Fig 2. However, we did add them to Table S1.

Additionally, we have added a sentence regarding the power of markers.

Added sentence: “It may also be the case that our low pig ancestry results suggest a scenario that hybrids in this study were probably the offspring of one or more backcross generation of pure-wild boar, which would halve the number of introgressed pig alleles at every generation. **While the selected STR markers consistently detected third backcross (BC2) generation hybrids (about 88 %, see electronic supplementary material, Table S2), it is feasible to assume a fourth or more backcross generation hybrid would be beyond the marker’s detection ability.**”

Table C Evaluation of power for our markers. Looking at F1, first backcross generation (BC1), second backcross generation (BC2), and third backcross generation hybrids (BC3).

Pig ID	Probability of F1, BC1, BC2, BC3 have only shared alleles*			
	F1	BC1	BC2	BC3
K104	0.000	0.012	0.128	0.369
K109	0.000	0.006	0.092	0.314
K110	0.000	0.012	0.128	0.369
K155	0.000	0.009	0.112	0.346
K156	0.000	0.008	0.104	0.335
K157	0.000	0.009	0.112	0.346
K175	0.000	0.008	0.105	0.335
K176	0.000	0.011	0.121	0.358
K183	0.000	0.008	0.105	0.335
K184	0.000	0.011	0.120	0.357
Average	0.000	0.009	0.113	0.346
SD	0.000	0.002	0.011	0.016

*The power of STRUCTURE might be better due to the consideration of allele frequency of pig and boar shared alleles

The other explanation that I can imagine, that would be really interesting to discuss (but I suspect speculative!) is that much of the recent hybridization, aka the hybridization in the last 10 years, is usually between male pigs and female boars, so there is no mtDNA introgression in the nuclear hybrids detected. This wouldn't explain why the mt-hybrids weren't detected using STRs, except, as the authors noted, that could be related to very old hybridization, or, as above, could have only happened at the initial hybridization event 10 years ago, and thus is beyond the detection ability of the 24 STRs. I'm not sure, but it could be cool to explore this idea. The manuscript notes 'random mating tendencies' (line 339), but this is just based on relatedness, isn't it? Is there assortative mating for species? If male pigs are more successful than female pigs, or if a huge proportion of domestic pigs died in 2011 (I can imagine they didn't fare well without being fed), perhaps the piglets of female boars who had pig sires did better than the piglets of female pigs x male boars (which would be carrying the pig mtDNA, as well as substantial nuclear DNA).

In either case, I think that the lack of concurrence between the nuclear and mt hybrids ought to be addressed. I think it's either a lack of power from the STRs, or an interesting biological mechanism that could only be suggested once a lack of power was ruled out.

Yes, we agree with the Referee 3 that this idea or possibility should be discussed. First, in the case of boar, assortative mating is likely not to occur. It is known well that there is no reproductive barrier between boar and pigs. That is the reason why boar and pig hybrids are often observed in the world, and the hybrid boar x pig is a global problem.

Additionally, after addressing the power of the STR markers, we have revised a paragraph in order to discuss the possibilities of lack of concurrence between the nuclear and mt hybrids.

Revised paragraph: “Despite such a sudden and large invasive force from pigs, our data likely suggest that most pigs failed to naturalize in the wild and survival of male pigs may have been higher. Here, we observed minimal selection of the pigs’ invasion from the remaining invasive pig mtDNA haplotype in wild boar and the reduction of pig alleles, which is more pronounced in the STR loci because they are a mixture from both sexes. However, our results also detected only two individuals as hybrids by both the mtDNA and the STR markers. Previous studies have also pointed out inconsistencies in hybrid determination by mitochondrial and nuclear markers [15]. Given that our selected markers consistently detected third generation backcrosses (see electronic supplementary material, Table S2), we can speculate that female pigs and maternal offspring had difficulty surviving. If pig farms tended to raise more female pigs than male pigs, then this survival bias may explain the lack of concurrence in hybrid determination by mitochondrial and nuclear markers, and the seemingly unnatural distribution of Q_1 values in mitochondrial hybrids. Additionally, in the U.S., male wild pigs tended to have higher survivability contributed to larger mass at birth [45]. Thus, it may be possible that male wild pigs or their paternal lineages in this area may have been more successful than female pigs.”